# Beyond Platform Economy: A Comprehensive Model for Decentralized and Self-Organizing Markets on Internet-Scale

**Mirjana Radonjic-Simic [1,*,†]** and **Dennis Pfisterer [2,†]**

[1] Baden-Wuerttemberg Cooperative State University, 68163 Mannheim, Germany

[2] Institute of Telematics, University of Lübeck, 23562 Lübeck, Germany; pfisterer@itm.uni-luebeck.de

[*] Correspondence: mirjana.radonjic-simic@dhbw-mannheim.de; Tel.: +49621-4105-1373

[†] Current address: Business Informatics, Baden-Wuerttemberg Cooperative State University, 68163 Mannheim, Germany.

**Abstract:** The platform economy denotes a subset of economic activities enabled by platforms such as Amazon, Alibaba, and Uber. Due to their tremendous success, more and more offerings concentrate around platforms increasing platforms' positional-power, hence leading towards a de-facto centralization of previously decentralized online markets. Furthermore, platform models work well for individual products and services or predefined combinations of these. However, they fall short in supporting complex products (personalized combinations of individual products and services), the combination of which is required to fulfill a particular consumer need, consequently increasing transaction costs for consumers looking for such products. To address these issues, we envision a "post-platform economy"—an economy facilitated by decentralized and self-organized online structures named Distributed Market Spaces. This work proposes a comprehensive model to serve as a guiding framework for the analysis, design, and implementation of Distributed Market Spaces. The proposed model leverages the St. Gallen Media Reference Model by adjusting existing and adding new entities and elements. The resulting multidimensional and multi-view model defines how a reference Distributed Market Space (a) works on the strategic and operational levels, (b) enables market exchange for complex products, and (c) how its instances might unfold during different life stages. In a case study, we demonstrated the application of our model and evaluated its suitability of meeting the primary objectives it was designed for.

**Keywords:** platform economy; distributed market spaces; reference models; smart city

## 1. Introduction

The fast development of the Internet and the related technologies initiated uniquely new and unprecedented opportunities for the development of new business models [1,2]. By leveraging technology, companies e.g., Amazon, Alibaba, Airbnb, and Uber created structures that enabled a wide range of human activities over the Internet. These online structures open up the way for radical changes in the way we work, socialize, and create value in an economy [3]. Such online structures are well-known as platforms and form the basis for the "platform economy"—a subset of social and economic activities enabled by online platforms [4].

Amazon, Alibaba, Airbnb, and Uber are examples of prominent and tremendously successful online platforms (for detailed information regarding the economic value of platform businesses, see, e.g., "Platform Companies in the Data Analysis 1995-2015" [2]). Even though they pursue different strategies, have different functions, and operate in different domains, each of them merely connects different user groups and enables them to exchange value conveniently and reliably [1,5]. At the core of

such success is that they managed to leverage technology and the economies of network effects in a way as to grow large value ecosystems and gain unprecedented power [3,6]. This power derives from their intermediary position and ability to leverage a massive amount of data collected from the interactions and transactions within their ecosystems. Even though platform ecosystems encompass networks of users, which are distributed and not under the direct control of the platforms, those networks are organized and orchestrated in a centralized manner [5]. As a result, the information processing and aggregation necessary for the functioning of these networks are centralized and exclusively controlled by the underlying platforms. This underpins the positional power of platforms, putting them in a position of a monopoly [5], where they can dictate the rules, control access, and thus lead to a de-facto centralization of previously decentralized offerings on the Internet. More and more initiatives, therefore, call for re-decentralization of the World Wide Web (e.g., SOLID Project by Berners-Lee), and hence for the re-decentralization of the Internet as a global market space (e.g., OpenBazaar [7]).

Another issue related to the modern platform economy is that platforms work well for individual products and services, but fail short in supporting consumers looking for complex products to satisfy a particular need. Complex products are personalized combinations of individual products or services that need to fulfill consumer-defined criteria and preferences [8,9]. Consequently, consumers looking for complex products, have to know where and how to find the optimal product/service combination, aggregate all relevant information manually, and put them in the context of personal preferences and requirements. This complexity of finding personalized product/service combinations causes frictions and increases transaction costs for consumers looking for such products.

To address the identified issues, we propose a "post-platform economy"—an economy that shifts the power from platforms to consumers and providers as the primary drivers of market exchange. It recognizes the primacy of consumers and their personalized demands enabling everyone and everything connected to the Internet to contribute to satisfying such personalized demands, being a consumer or a provider, or both at the same time. Furthermore, in a post-platform economy (as detailed in Section 2.1), consumers and providers are considered equal in their rights and responsibilities as they can engage in complex product scenarios directly without any intermediaries. As participants, they are constitutive parts not only through the intention to participate in market exchange but also through their intention to provide for the underlying market mechanisms. Accordingly, we define post-platform as a set of economic activities enabled by self-organized and strictly decentralized online structures named distributed market spaces. The primary purpose of distributed market spaces is to counter the adverse effects of growing platform-power and lower transaction costs for complex products while maintaining the benefits and enabling nature of the centrally organized solutions.

In this work, we present our reference model for Distributed Market Spaces—a comprehensive model that serves as a guiding framework for the analysis, design, and implementation of decentralized and self-organized online structures to facilitate the emergence of the post-platform economy.

Our model leverages the St. Gallen Media Reference Model [10,11] by adjusting existing and adding new entities and elements required to meet the design objectives of distributed market spaces. The resulting multidimensional and multi-view model defines how a reference distributed market space (reference DMS)

- works on the strategic and operational level,
- enables market exchange for complex products,
- and how its instances might unfold during different life stages.

The applicability of the proposed model is demonstrated with a case study in the context of a Smart City project. The findings of the conducted case study suggest that the proposed model can be beneficial in two ways. Firstly, it can provide insights essential for understanding different aspects, entities, and elements of self-organized and decentralized online structures. Secondly, it can assist as a guideline on how to design and implement instances of distributed market spaces for a specific application context. We, therefore, believe that our proposed model contributes to the aforementioned

re-decentralization initiatives, as it guides market participants (consumers and providers) to establish and enhance market spaces on their own in which they can engage in market-exchange of complex products directly and reliably.

This work is structured as follows: Section 2 provides an overview of the concept of distributed market spaces and theoretical backgrounds to reference modeling. Next, Section 3 examines the related work and identifies useful elements for modeling of distributed market spaces. Section 4 presents our proposed model describing its dimensions, views, phases and stages, and specifying their core elements. Thereafter, Section 5 demonstrates the application of the proposed reference model with a case study, followed by the discussion of the key findings in Section 6. Section 7 concludes this work with a summary and an outlook to future work.

## 2. Background and Approach

This section provides theoretical backgrounds to this work. It first introduces the concept of distributed market spaces describing its key characteristics and primary objectives. Afterward, it provides an overview of reference modeling and outlines the approach we used to develop this work.

### 2.1. Distributed Market Spaces-Characteristics and Objectives

The primary purpose of distributed market spaces is to facilitate the emergence of the post-platform economy, and thus

- to alleviate the adverse effects of rising platform-power, and
- lower transaction costs for complex products

while maintaining the benefits and the enabling-nature of contemporary platform models. Given that, the key drivers of distributed market spaces are decentralization and novel consumer orientation, whereby novel consumer orientation emphasizes personalized consumers' needs represented by the market exchange of complex products.

#### 2.1.1. Decentralization as a driver

At the core of the platform business model is the creation of value by providing an open, supporting infrastructure that enables consumers and providers to plug in to interact and transact with each other [3,12]. The anatomy of a platform model is illustrated in Figure 1, on the left.

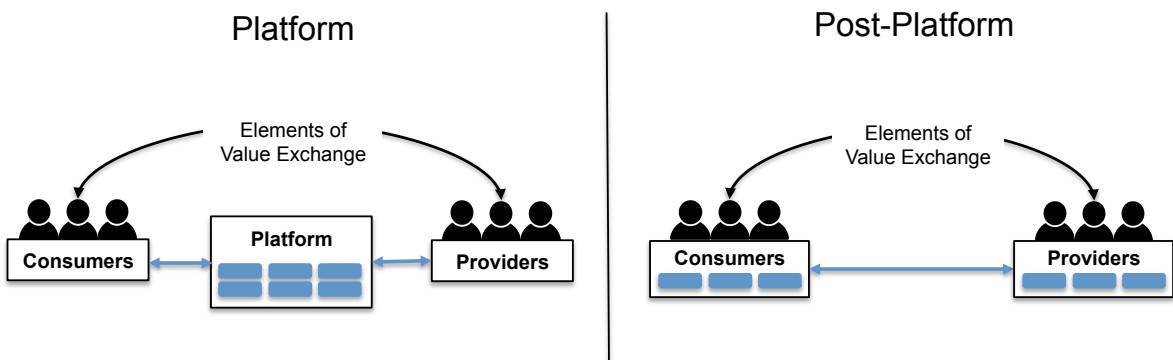

**Figure 1.** Platform economy vs. post-platform economy.

The supportive infrastructure (i.e., the platform) provides core functions related to decision-making, information processing, and governance of the network built around this infrastructure. Core functions are under the direct control of platform owners and build a collection of mechanisms, through which they influence and exercise control over platforms' participants [13]. As a

result, more and more economic activities in the platform economy are orchestrated through centrally created and managed networks [2,5]. As aforementioned, this underpins the positional power of the platform, putting them in a position of a monopoly position, where they can dictate the rules, control access, and offerings, and thus lead to de-facto centralization of previously decentralized offerings on the Internet. To alleviate such effects, distributed market spaces are required to shift the paradigm of contemporary platforms towards decentralization and the provision and support of core functions in a decentralized manner. This shift implicates that core functions need to be provided by the market participants themselves, participants who self-organize to ensure that core functions are provided collaboratively (see Figure 1, on the right).

Consequently, the concept of distributed market spaces needs to direct towards decentralized decision-making, information processing, and governance. Therefore, we characterize *distributed market spaces as strictly decentralized and self-organized online structures*. *Decentralized* refers to aspects of decision-making and information processing, and *self-organized* characterize decentralized governance. Furthermore, *self-organized* underlines the decentralized character of distributed market spaces and emphasizes its ability to empower participants to establish and uphold an exchange environment on their own, where they can exchange value reliably and directly.

### 2.1.2. Complex Products as a Driver

Complex products refer to arbitrary combinations of individual products and services that need to fulfill a particular consumer-defined context in order to fulfill a personalized need. Consider, for example, a couple who wants to spend a pleasant evening with friends at the theatre. As a consumer, this couple demands a combination of services that includes: tickets for the theatre, reservation of a table at an Italian restaurant, finding parking close to both locations, and engaging a well-rated babysitter to watch after their children. The demand spans four different service domains (i.e., ticketing, gastronomy, parking, and babysitting) and has to consider contextual information regarding the schedule, location, and ratings of a particular service.

For each of these domains, several platforms exist. However, they focus only on products and services from the supported domains (e.g., Eventim (http://www.eventim.com) for domain ticketing and MyTable (http://www.mytable.com) for gastronomy), which is why platforms are often called "verticals". Figure 2 illustrates the vertical orientation of contemporary platforms spanning over domains ($Domain_1, \ldots, Domain_n$).

For our couple, it implies that it needs to combine different verticals in order to get the demanded service combination, put it all together, and all of this in the context of schedule and location. This requires a high level of personal involvement and manual activities, which contradicts the primary purpose of platforms as matchmakers [12]–connecting the right consumers with the right providers and by doing so, significantly reducing transaction costs. Thereby transaction costs include any costs incurred in participating in interactions and making a market exchange [14].

Accordingly, the concept of distributed market spaces shifts the vertical orientation of the platforms towards consumers and their increasingly horizontally oriented complex demands. Figure 2 indicates this shift and complex products ($CP_1, \ldots, CP_n$), as shown on the right. A further main characteristic of distributed market spaces is hence the capability *to empower consumers* to formulate their complex demands, and based on that, *to facilitate the market exchange of complex products*–in the same effective way as platforms support individual products and services today.

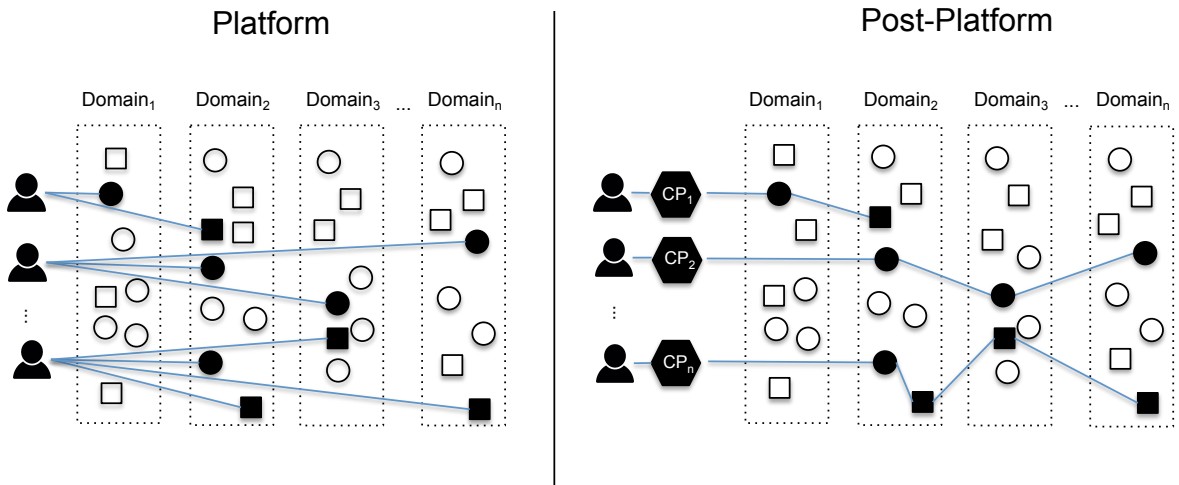

**Figure 2.** Complex products in platform vs. post-platform economy.

As to consideration above, the objectives identified as primary design goals that distributed market spaces as decentralized and self-organized online structures need to meet are the following:

- Objective 1—Facilitating decentralization and openness.
- Objective 2—Enabling market transactions of complex products directly and reliably.
- Objective 3—Supporting scalability and allowing the cross-domain market exchange.

*2.2. Reference Modeling*

Reference models are generic conceptual frameworks that represent a domain or a class of domains [15,16]. Reference models formalize recommended practices for the focus domain [17], and in this manner, they provide shared models that facilitate learning and lead to a better understanding of underlying domains.

As in [15], there are three common characteristics of reference models: best practices, universal applicability, and reusability. Best practices indicate that a reference model for a particular domain provides best practices. Universal applicability indicates that a reference model is valid for a class of domains, and reusability that they can be understood as blueprints for information systems development and thus could be reused in a multitude of different information systems projects.

The literature on reference modeling suggests that there are basically two approaches to reference modeling. The first approach is based on observing many instances available in practice and extracting common elements into reference models. This approach is suitable when a sufficient number of relevant instances for the particular domain are available. The second approach addresses the opposite case when the underlying domain is not researched enough, but still, similarities can be drawn to other reference models. It is based on the leveraging and adjusting of existing reference models in order to meet the objectives and contextual requirements of the underlying domain.

As distributed market spaces are a new concept, the second approach was applied. The reference modeling process for the domain of distributed market spaces encompassed three steps:

- Step 1—In Section 3, we reviewed the relevant reference models, with the focus on applicability, reusability, and adaptability for the modeling context of distributed market spaces.
- Step 2—In Section 4, we leveraged and adjusted the identified models and entities, adding additional entities and elements to meet the stated objectives of distributed market spaces.
- Step 3—In Section 5, we demonstrated the application of the proposed reference model in the context of a case study and evaluated its suitability of meeting the objectives it was designed for.

## 3. Related Work

This section examines existing reference models relevant to the modeling context of distributed market spaces. The modeling context is determined by the prerequisite discussion and objectives stated in Section 2.1. Accordingly, existing reference models for market-oriented environments are reviewed and classified by applying the framework by Braun and Esswein [18] to three main classes:

- Practitioner Reference Models.
- Scientific Business Process Reference Models.
- Scientific Multi-View Reference Models.

Each of these three reference model classes has been examined with the focus on its applicability, reusability, and adaptability:

- Applicability—to assess the extent to which they integrate concepts and models required for reference modeling of distributed market spaces.
- Reusability and adaptability–to assess how their elements can be reused and adapted to be of service for the distributed market spaces reference model.

### 3.1. Practitioner Reference Models

Reference models of this class (e.g., [3,5,6,12,19]) describe platforms as online structures, which are often implemented in practice and proven very successful in facilitating market exchange over the Internet. Moazed and Johnson [5], Parker et al. [3], and Evans and Schmalensee [12] propose models that focus on the anatomy of a platform that consists of four main building blocks: the audience building, matchmaking, providing tools and services and setting rules and standards. The Rocket Model [6] introduced by Reillier and Reillier uses a similar structure, adding the data-driven optimizing block considered essential for the strategic development of platforms as ecosystems. Additionally, the model by [6] also considers reference modeling in the context of the platform life-cycle. It introduces a four-stage model distinguishing between the stages pre-launch, ignition, scaling-up, and maturity. The PIK model (Platform Innovation Kit [19]) goes in the same direction and offers a comprehensive toolset for the extraction of platform models for a particular application scenario. PIK proposes nine canvases with pre-defined questions to support the modeling process. The canvases cover concepts and elements ranging from environment scanning and describing application scenarios over ideation and defining value proposition and designing of required platform services, to the strategy definition for long-term development.

*Applicability:* The main advantage of practitioner reference models is their recommendation character as they provide solid guidelines to other practitioners looking to conceptualize and instantiate platform models for specific application scenarios. However, none of these models makes recommendations about underlying systems and technology-related aspects required to support platforms on the operational level. Concerning their applicability, reviewed practitioners reference models fall short of serving as a reference model for distributed market spaces as they are single-sided and biased to the perspective of the platform owner. As such, they assume centralized ownership structures and fail to include concepts and elements that support aspects of self-organization and self-governance. Initiatives and projects around platform cooperativism by [20,21], collaborative economy [22], or similar earlier research towards open cooperativism, e.g., [23] can assist in addressing these concepts. Yet, they do not explicate elements of self-organized and self-governed models but contribute to the conceptual perspective of such structures.

*Reusability and adaptability:* Assessed practitioner reference models provide at least two elements considered useful and adaptable for distributed market spaces. On the one side, this is a matchmaking service that facilitates market transactions in a way that is proven remarkably beneficial for creating efficiencies and for building communities and establishing positive network effects ([3,5,12]). On the

other hand, the concept of platform life-cycle by [6] can be used and adopted in a way to distinguish between different perspectives and objectives related to distinct life-stages of distributed market spaces.

### 3.2. Scientific Business Process Reference Models

Reference models summarized in this class (e.g., [24–27]) focus on the definition of business processes required to support market transactions on the Internet. The H-Model by Becker and Schutten [24] introduces a reference model for electronic retail. It describes tasks relevant for the modeling of different parts of a retail enterprise, grouping them in three perspectives (i.e., viewpoints): business functions, processes, and static data. Similar to the H-Model, the E-MEMO reference model [25] by Frank introduces a comprehensive library of process-models for e-commerce. The E-MEMO reference model divides process models into different categories based on the business function they cover. These categories range from pre-sales communication over the initiation, pricing, order processing to customer services related to after-sales. While the aforementioned models [24,25] emphasize the inter-organizational processes, Kollmann offers a model that specifies intra-organizational processes of electronic marketplaces (i.e., e-marketplaces). Kollmann's model defines processes around the phase model of market transactions divided into the phases of information, negotiation, settlement, and after-sales. It further suggests three different views on defined phase-related processes: the view of a consumer, the view of a provider, and the view of an intermediary. The intermediary view is thereby considered substantial as an intermediary is considered responsible for providing the underlying infrastructure and integrating processes necessary to facilitate market transactions on an e-marketplace.

*Applicability:* Even though these models provide very detailed guidance on business functions and the related process of an exchange environment, their applicability is limited as they only support the prevailing paradigm of intermediated models with strictly separated roles consumers, providers, and intermediaries. Moreover, business process reference models fail short of providing any guidelines on how these processes need to be supported by an underlying system. The model by [27] addresses these issues by introducing a layered E-Commerce Reference Architecture (ERA). ERA proposes processes grouped into three layers: business, application, and technology layer, as well as processes that define the relationships among business functions across these three layers. Nevertheless, as with the other aforementioned models, the ERA model failed to integrate strategy-related processes showing how an exchange environment needs to be organized and modeled on the strategic level to facilitate market exchange.

*Reusability and adaptability:* Business process reference models provide relevant market transaction processes ([26,27]) in particular, phase-related interaction processes can be adapted in a way to support the specificities of complex products. Especially useful are recommended processes that describe interactions among market participants (consumers and providers) related to return and refund, review, and dispute resolution processes.

### 3.3. Scientific Multi-View Reference Models

The Multi-View reference models summarize conceptual frameworks for the design of market-oriented networked structures that incorporate multiple views (e.g., [10,11,28,29]). The reference model proposed by Menasce [28] introduces a four-layer reference model for electronic business. It is composed of a business model, a functional model, a customer behavior model, and an IT resource model. The reference model for collaborative networked organizations (CNO) [29]) follows a comparable approach proposing structural, componential, functional and behavioral models, but in contrast to Menasce 's four-layer reference model, the CNO emphasizes the collaboration aspect and integrates models for the design of environments organized around and based on collaborative networks. Although the CNO reference model provides solid foundations on how to conceptualize business environments based on collaborative networks, it fails to integrate elements considering

the system and infrastructure view as is the case with the previously discussed other two reference model classes.

The St. Gallen Media Reference Model (MRM) [10,11] addresses these shortcomings by introducing a two-dimensional framework for the conceptualization of reference media for electronic markets. The term "media" is related to the concept of a platform as a communication space built for "social interactions, which allow the participants to meet and which embed them in a common physical, logical, and socio-organizational structure" [10]. The horizontal dimension of the MRM represents the market transaction model comprised of knowledge, intention, contract, and settlement phases, and the vertical dimension groups the four views. In MRM views are organized into four layers: community, interaction, services, and infrastructure. The community view describes the platform participants, their roles, and the organizational structure defining the relationships among these roles together with their obligations and rights. The interaction view refers to the relevant processes and builds upon the underlying services. The service view comprises all services in the four market transaction phases that need to be available on the platform, and the infrastructure view represents the information and communication infrastructure required for the implementation of the service view.

*Applicability:* The MRM has been successfully applied in many different domains (e.g., m-commerce [30], collaborative networks [31], service systems [32], enterprise mashup environments [33], and marketplaces for cloud services [34]. Regarding its applicability in the context of this work, the MRM is considered suitable to be leveraged as the theoretical framework for the development of a new reference model. The rationale behind is two-fold: Firstly, it approaches the design process from different viewpoints taking into considerations strategic, operational, service and infrastructure concerns of an exchange environment, and secondly, it integrates these views with the underlying market transaction model, in order to facilitate market exchange over such environments.

*Reusability and adaptability:* For a reference distributed market space, the conceptual structure of MRM (both dimensions views and phases) needs to be adjusted and modified to cope with the specificities of complex products. Moreover, the MRM reference model only considers the design stage of a market-oriented structure and, therefore, needs to be extended with an additional dimension in order to acknowledge different life stages of self-organized and governed online structures.

*Closing remarks on related work:* Regarding the consideration above, each of the examined reference models classes contain elements considered useful for reference modeling of distributed market spaces. However, neither of the examined models can serve as the reference for the design and conceptualization of distributed market spaces. Therefore, a novel multi-view reference model is required–a comprehensive model that on the one side utilizes existing models using them as theoretical backgrounds, and on the other, proposes new concepts and integrates new elements to meet the objectives of distributed market spaces during their life cycle.

## 4. Proposed Reference Model for Distributed Market Spaces

The reference model for distributed market spaces is designed and developed to serve as a framework to structure the analysis, design, and implementation of distributed market spaces as self-organized and decentralized online structures to facilitate the emergence of the post-platform economy.

The proposed model extends the St. Gallen Media Reference Model by Schmid and Lindemann [11] adjusting existing and introducing new elements to meet stated objectives (cf. Section 2.1). The changes refer to the following:

- The vertical dimension (*Views*) was modified with an ecosystem view to integrate the ecosystem perspective of self-organized and governed structures. Further services are introduced to implementing the ecosystem view and related interaction processes;
- The horizontal dimension (*Phases*) was extended by a new phase to integrate additional activities necessary for the market transactions of complex products to realize;

- A new dimension (*Stages*) was added to integrate different concerns in different life stages of distributed market spaces.

Figure 3 presents the reference model for distributed market spaces. It spans three dimensions defining a reference distributed market space (*a reference DMS*) in the dimension of *Views, Phases*, and *Stages*.

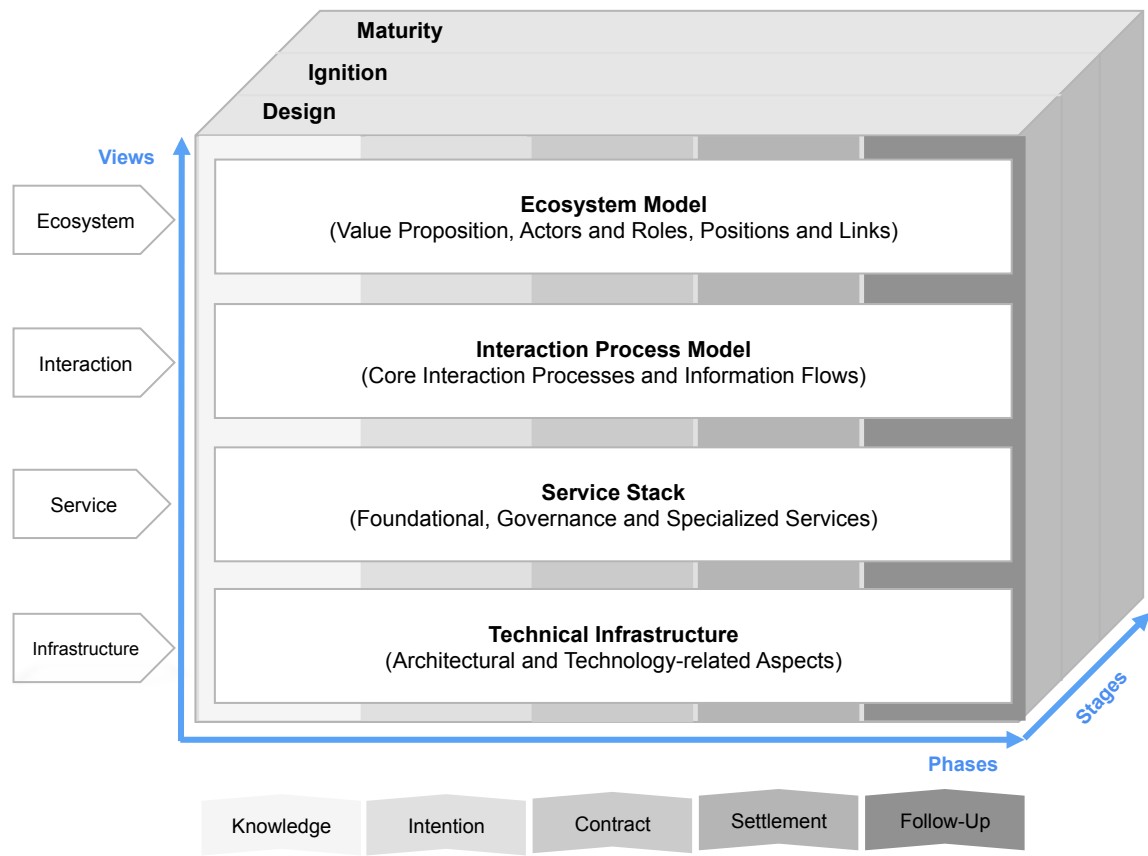

**Figure 3.** Reference Model for Distributed Market Spaces.

The *Views dimension* describes a reference DMS taking four different points of view:

- The Ecosystem view maps the organizational structure of a reference DMS as an ecosystem. It proposes an ecosystem model that outlines actors, their roles, and primary activity flows and explicates how identified actors and activities need to link and align in order for the ecosystem's value proposition to materialize.
- The Interaction view specifies the core interactions among identified actors taking different roles at the operational level of a reference DMS. It proposes an interaction process model that specifies the interaction processes, relevant activities, and resulting information flows required for market transactions of complex products through a reference DMS.
- The Service view defines services that a reference DMS must provide to its participants in order to facilitate the ecosystem and interaction view. It introduces a service stack that implements the ecosystem model and its core interaction processes specified by the interaction phase model.
- The Infrastructure view describes the technical infrastructure of a reference DMS for the implementation of the service view. It considers architectural and technology-related aspects and forms the basis for the implementation of the defined service stack.

The *Phases dimension* defines a reference DMS as a market-oriented environment for supporting transactions of complex products. As shown in Figure 3, it is based on a phase model for market

transactions of complex products, and it comprises the knowledge, intention, contract, settlement, and follow-up phases. The purpose of the phase model is to define how to initiate, arrange, and settle contractual agreements for market transactions of complex products in the most efficient manner. In this case, efficient manner refers to lowering transaction costs for consumers looking for transactions of complex products directly and reliably.

The *Stages dimension* comprises the life stages a reference DMS might undergo during its development and growth. As shown in Figure 3 these are the design, ignition, and maturity stages. Each of them focuses on different concerns and therefore has different priorities. While the main priority of the design stage is to blueprint, prototype, and launch a DMS for a specific application domain, the priority of the ignition stage is to build the critical mass of participants as the prerequisite for the maturity stage. If the maturity stage is achieved, the main priority could be to retain existing participants (network) and connect to others for sustainability and growth.

Together the dimensions *Views, Phases*, and *Stages* build *a comprehensive Multi-View Reference Model*, which describes how a reference DMS works on the strategic and operational level, enables market transactions for complex products and, how its instances might unfold during different life stages.

### 4.1. Phase Model of Market Transactions for Complex Products

The phase model of market transactions for complex products defines the necessary interactions between market participants, consumers, and providers engaged in transactions of complex products. Hence, it lays the ground for lowering transaction costs for consumers looking for transactions of products over DMS.

The proposed phase model enhances the existing market transaction model as used in MRM [10,11]. The extensions refer to

- The integration of the 'Follow-Up' phase, a new phase that integrates interaction processes among market participants that happen after the settlement.
- The integration of additional processes that address the specifics of complex products (cf. Section 2.1)

As a result, the phase model of market transactions for complex products encompasses five phases: *Knowledge, Intention, Contract, Settlement*, and *Follow-Up*.

Each of these phases represents a group of activities by involved participants, and each of them has a defined output or a phase result. Figure 4 presents the proposed phase model and summarizes the results of each of the five phases, as described in the following.

- In the *Knowledge phase,* market participants acquire an overview of the supply and demand in a distributed market space. Providers publish their offers by publishing descriptions of the products and services they offer. Consumers formulate their demands as complex product requests and, based on that search for potential providers that can provide parts of the required complex product. Consequently, the knowledge phase ends with a product/service description and a formulated complex product request accompanied by a list of possible transaction partners.
- In the *Intention phase,* market participants negotiate conditions for an agreement for the particular complex product. It covers the process of sending "Requests for Offer" (RfO) to the potential transaction partners (consumer), and providers send back particular offerings in a way including the price tag, payment mode, and delivery conditions. Consumers then aggregate all received offerings, and create complex product proposals, which they rank based on the defined requirements and constraints. Consumers might then select one complex product proposal, which best suits their demands. The chosen complex product proposal (consumer side) and offerings (provider side) represent the phase results. They are the starting point for the next phase and form the basis for the contractual agreement to be made.

- In the *Contract phase,* consumers and provides substantiate the negotiated agreement represented by a legally binding contract. From the consumer side, a complex product contract is considered an umbrella contract since it incorporates different arrangements for different parts of the complex product. The umbrella contract represents a one-to-many contract situation and requires consumer's involvement in several contractual processes (one for each product or service). On the provider side, the contracting process is regarded as a one-to-one contract situation with an additional activity regarding the confirmation of a pending contract. The phase ends with an agreed legally binding complex product contract, which is the starting point for the settlement phase.
- The *Settlement phase* serves to fulfill the obligations resulting from the complex product contract agreed in the contract phase. Similarly to individual products and services, the settlement phase of complex products encompasses interaction processes related to delivery, payment, and logistics. Depending on the type of the exchanged product or service as well as the involved providers, the settlement phase might include additional sub-processes related to the type of settlement, which will be detailed in the following.
- The *Follow-Up phase* completes the market transaction for complex products. As the fifth phase, follow-up supports interactions between transaction partners that happen after the settlement. These are interactions and activities related to reviews of settled transactions, customer support, management of return and refund as well as management of disputes among transaction partners.

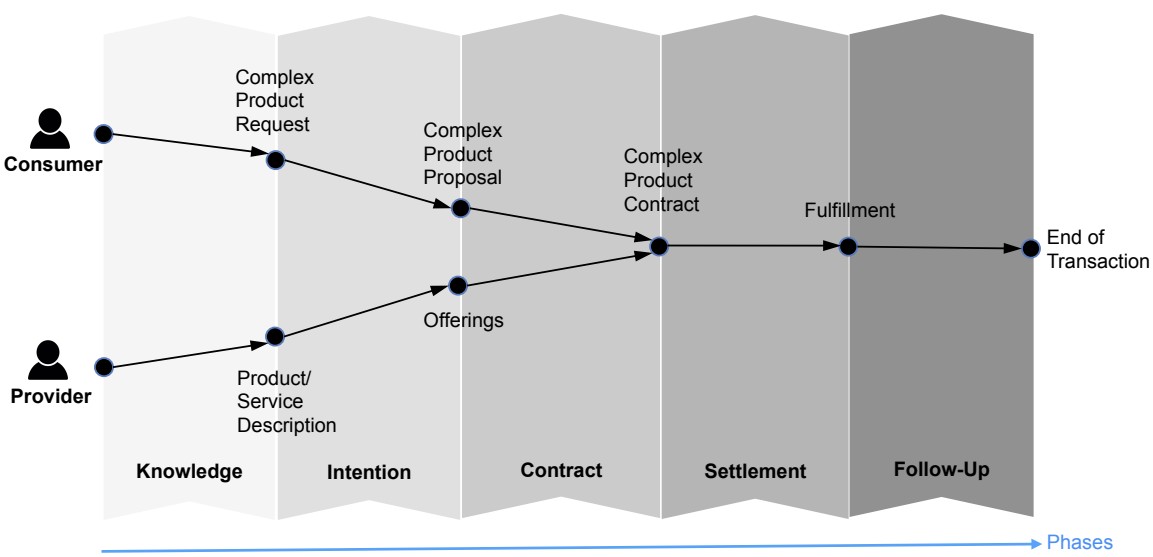

**Figure 4.** Phase model of a market transaction for complex products.

The proposed phase model of market transactions for complex products is elaborated on in more detail in Section 4.3. In this section, the inner workings of each phase, their processes, and related activities are detailed and presented in the context of the interaction view of a reference DMS.

*4.2. Ecosystem View*

The ecosystem view maps the ecosystem structure of a reference DMS. The *Ecosystem Model* blueprints the proposed ecosystem structure by outlining the primary activities, actors and their roles, and how actors and activities need to link and align in order to support the shared purpose of the DMS ecosystem.

Ecosystems are economic communities supported by am underlying interacting organizations and individuals [35] that use common standards and collectively provide goods and services [36].

As such, ecosystems consist of numerous loosely interconnected actors who create value through the process of streichen and competition [37,38]. This explicit dependence of involved actors who rely on one another is considered the essential feature of ecosystems that distinguishes them from other interconnected environments, e.g., value networks or value chains.

As organizational models, ecosystems are defined by two primary characteristics: Firstly by how value is created, and secondly by how it is shared in order to satisfy the individual and collective motivation of actors participating in the ecosystem [37–40]. Consequently, ecosystem models are considered constructs composed of entities and elements required to specify how value is created and shared among participating actors.

Literature provides various approaches and concepts in order to formalize ecosystem models e.g., BEAM [38], MOBENA [41], 6c [39], VISOR [40], Value Design [42], and Ecosystem Construct [43]. Even though each of these modeling approaches has its initial focus, they address ecosystem modeling from different perspectives [43]:

- Ecosystem-as-affiliation: views ecosystems as communities of associated actors defined by their network affiliation and as a complement to a focal actor [38–41].
- Ecosystem-as-structure: views ecosystems as alignment structures of activities and actors defined by a shared value proposition, rather than being an affiliate of a focal actor [42,43].

For the modeling of an ecosystem for a reference DMS, the ecosystem-as-structure perspective applies, and consequently, the modeling approach by [43] concerns. The rationale lays in the definition and requirements of a reference DMS. As a self-organized and governed structure of actors with equal rights and responsibilities, the DMS ecosystem needs to organize in a way to allow the shared value proposition to realize in a decentralized manner. As there is no focal actor, this requires that actors align following an agreement on how value is created and shared within the ecosystem they constitute. Such an alignment, thus, refers not only to shared motivation and incentives as the case with ecosystems-as-affiliation but also requires actors' consistent engagement. As will be discussed below, part of consistent engagement is the commitment to take different roles and provide resources and services to uphold the ecosystem.

Figure 5 presents the resulting *Ecosystem Model* for the reference DMS. *Value proposition* defines the shared purpose of the DMS ecosystem and is formulated as an *end-user enabled ecosystem for the market exchange of complex products directly and reliably*. The following are the core underlying elements of the stated value proposition:

- Activities –defining the primary activity groups and discrete actions to be undertaken.
- Actors–specifying the entities that undertake these activities, taking different roles.
- Positions–specifying where in the flow of activities actors are located.
- Links–specifying how actors taking different roles need to interact and what value they need to exchange.

Core elements mutually depend on each other and together describe how value is expected to be created and shared within the ecosystem. Hence, they conceptualize a decentralized environment of interdependent collaboration that is the organizational structure of the DMS ecosystem underlying the stated value proposition.

**Activities**

There are three primary activity groups that the DMS ecosystem needs to support to realize the stated value proposition:

- Demand and Supply
- Market Transactions
- Ecosystem Foundation

*The demand and supply* activity group defines activities related to the composition and description of complex product requests on the consumer side (i.e., demand) and the description and publishing of products and services on the provider side (i.e., supply).

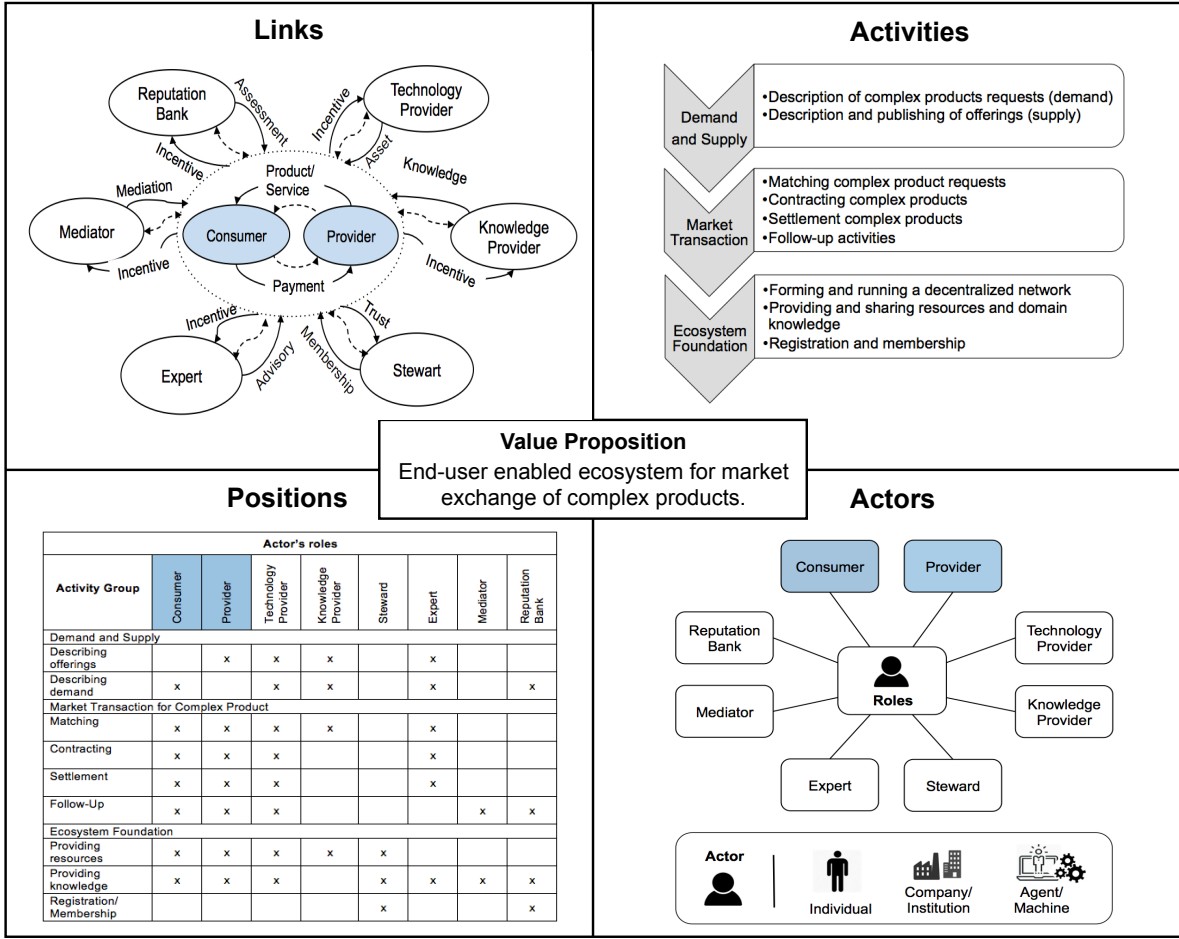

**Figure 5.** An ecosystem model of a reference DMS.

*The market transaction* activity group defines activities to support the phase model of market transactions for complex products. As previously described, these are activities necessary to support interaction processes in each of the phases of negotiation, contracting, settlement, and follow-up of complex products.

*The ecosystem foundation* activity group defines activities to build the foundation, essential for setting up and operating the ecosystem. It includes forming and running the network by providing resources and services (e.g., hosting, tools) and domain knowledge necessary for the market exchange in a specific domain (e.g., domain ontologies and vocabularies for that particular domain).

**Actors and Roles**

Actors in the DMS ecosystem can be everyone or everything connected to the Internet intending to engage in complex product scenarios. That includes individuals, companies, institutions or associations, and other networks, as well as autonomous actors such as software agents or machines. As shown in Figure 5, there are at least eight roles that actors can take:

- Consumer
- Provider
- Technology Provider
- Knowledge Provider
- Steward

- Expert
- Mediator
- Reputation Bank

*Consumer* and *Provider* are considered shaper roles as they shape the value proposition and, thus, the birth of the DMS ecosystem. The other roles (*Technology Provider, Knowledge Provider, Steward, Expert, Mediator,* and *Reputation Bank*) are enabler roles. Their purpose is to enable the ecosystem to provide comprehensive services to the shaper roles. Therefore the primary function of enabler roles is to enable the ecosystem's value creation by undertaking activities as mentioned above.

Roles are motivated by two sides: on the one hand side, they are motivated by shared purpose in order to realize the ecosystems' value proposition, and on the other side, they are driven by individual motivation. Shared motivation designates the commitment to be a constitutive part of the ecosystem's alignment structure and the continuous engagement in order for the ecosystem to uphold. Individual motivation refers to the additional value an actor expects from the participation in the DMS ecosystem. Such an expected value might differ from role to role, be subject to various actor types, and even change over different life stages of the DMS. Even though each of the roles has different functions and is responsible for different activities, roles can overlap and be assumed concurrently, as they do not exclude each other. For example, a consumer (shaper role) can also take the role of, e.g., technology provider, or an expert (enabler role) at the same time. Table 1 summarizes the abovementioned roles describing their functions and stipulating possible individual motivation or expected value from the participation in the DMS ecosystem.

### Positions

In order for the stated value proposition to realize, the definition of the necessary activities and the identification of actors who need to undertake these activities as well as the assignment of roles are necessary, but not sufficient. In order to create value, the ecosystem's actors in their different roles need to align around the activities and take a particular position in the overall value creation. Positions, as illustrated in Figure 5, provide an overview of where in the flow of activities the identified actors need to be located. Single roles can contribute to several activities, and specific activities might require several roles to engage. For example, in order to support consumers in formulating demand, that is, composing complex products as arbitrary combinations of individual products and services, several roles need yo be engaged. Besides the consumer who initiates the process, the technology and knowledge provider are required to provide tools and knowledge to enable the composition of complex products and the integration of the contextual information. Depending on the complexity and level of personalization, the composition of the complex products might also involve further roles. In the example, experts might support composing the required product/service combination, and the reputation bank might provide information about the reputation and worthiness of the possible providers. In that way, enabler roles are supporting consumers proceeding in a more targeted manner, narrowing the selection of the potential providers at the beginning of the market transaction, already contributing to a lowering of transaction costs.

### Links

Links illustrate how actors taking different roles need to interact and specifies the transfer between them. Figure 5 visualizes links in the form of a flow diagram, which outlines the overall pattern of exchanges within the DMS ecosystem. The focus lies on shaper roles, both for consumer and provider, and the visualization of the most important interactions with enabler roles and resulting exchanges.

The nodes represent actors performing a particular role and the arrows the essential interactions indicating the "value exchanged" between these roles. Solid lines denote the "tangible" exchanges such as product/service delivery or payment as is the case with consumer and provider role (see Figure 5). Dashed lines indicate additional exchanged value that is considered "intangible" like for example, feedback, reputation, or usage-related data. Regarding the interaction between shaper roles, this might be the contextual information a provider might receive from a consumer requesting an individual product or service. Such additional information is considered valuable as it can be used to

increase the contextualization of offerings, and thus, enables the provider to provide in the consumer context. Vice-versa, a consumer, can get personalized bundles of a product or service that best fits his demand based on the provided contextual information. Links explicate the overall value exchange within the DMS ecosystem necessary to realize its proposition and, thus, the shared motivation of the named roles. Further, links consider the content of transfers required to satisfy the individual motivation of role, i.e., the expected value from the participation in the DMS ecosystem (cf. Table 1). Note that Figure 5 presents only the most essential exchanges among the named roles.

**Table 1.** Actor's roles and their motivation/expected value [8].

| Role | Description | Motivation/Expected Value |
|---|---|---|
| Consumer | Looking for a complex product. | To satisfy personalized needs defined through market transactions of complex products directly and reliably. |
| Provider | Offering products or services in one particular domain or many domains. | To earn revenue per product/service sold (payment). To increase the visibility of offerings. To increase the level of customization based on contextual information provided by consumers. |
| Technology Provider | Providing technology assets (resource/tool/service) to support market transactions of complex products. | To contribute to the ecosystem foundation. To earn revenue by guaranteeing availability only to paying users (incentive). To leverage usage-data for improvement and developing new assets. |
| Knowledge Provider | Providing domain knowledge. | To contribute to the shared knowledge base. To earn revenue by providing paid knowledge-based services (incentive). |
| Steward | Registering the ecosystem's members after being granted access. | To contribute to the self-governance capability of the ecosystem. To ensure congruence between members, rules, and norms. |
| Expert | Offering expertise and advice to inform decision making. | To earn revenue through advisory and user's feedback (incentive). |
| Mediator | Offering mediation to support resolving disputes and conflicts. | To earn revenue through mediation and user's feedback (incentive). |
| Reputation Bank | Assessing ecosystem's members regarding their reliability, solvency, and worthiness. | To capture two-sided reviews about conducted transactions needed for a qualified assessment of members (assessments). To promote an adequate level of trust among the ecosystem's members. |

### 4.3. Interaction View

The interaction view specifies the core interactions required for market transactions of complex products via a reference DMS. The purpose of the resulting *Interaction Process Model* is to define the critical processes between shaper roles (consumers and providers) structured around the phase model of market transactions for complex products.

Figure 6 shows a high-level overview of the proposed process model. It presents the core interaction processes between a consumer looking for a complex product, and a provider (or many of them) engaged in the market transaction for that particular complex product. It summarizes the necessary processes for each phase and the resulting information objects, explicating their relationships and locations inside each of the phases of the market transaction model (cf. Figure 4). The high-level overview and all related sub-processes are modeled and described using the Business Process Model Notation (BPMN 2.0 [44]).

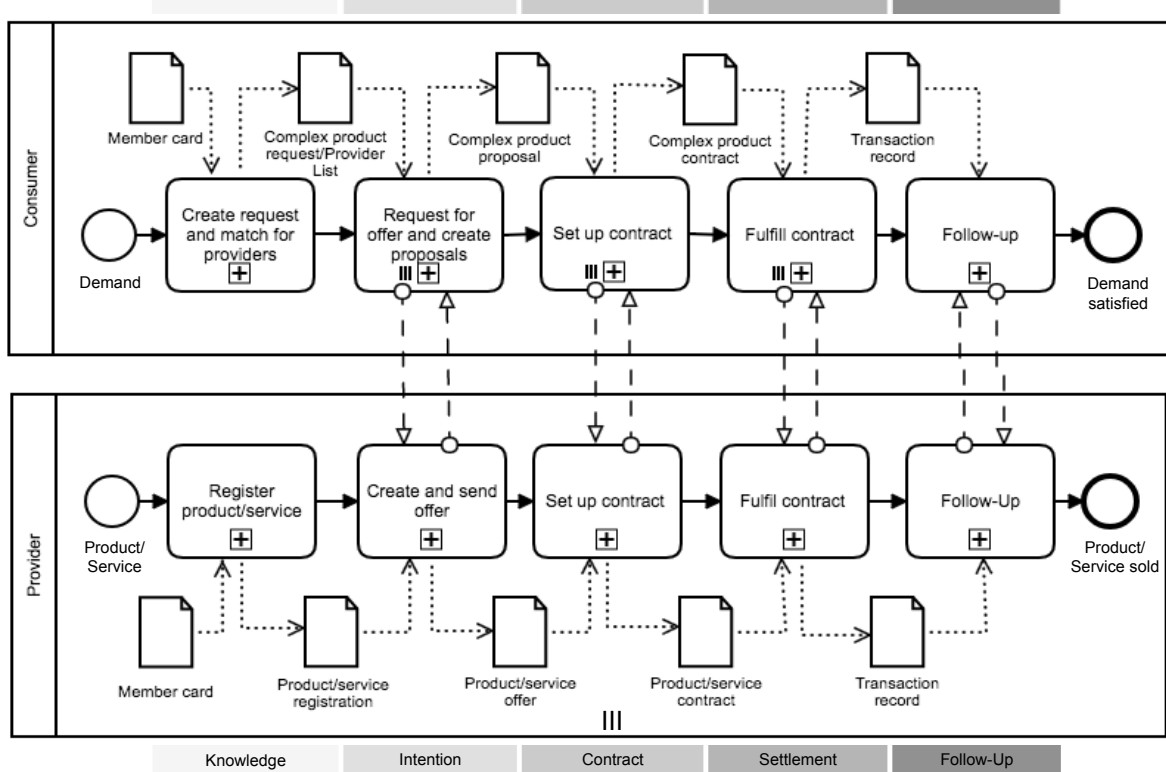

**Figure 6.** Interaction process model of a reference DMS—a high-level overview as a BPMN2.0 diagram.

As indicated in Figure 6, the high-level interaction process starts with the demand for a particular complex product on the consumer side and the idea for a concrete product or service on the provider side. The prerequisite for participation in the process is DMS membership. DMS memberships are represented by *Member cards*, which are the basis for all further process steps.

For consumers (see upper lane), these are process-steps, which enable formulating demand (*Complex product request*) and acquiring an overview of potential providers who might satisfy such demand (*Provider list*). For the providers (bottom lane), these are the process steps that provide for a description and registration of offerings (*Product/service registration*). The resulting information objects (*Complex product request/Provider list*) enable consumers to reach potential providers and thus start the interaction processes of the following phases:

- The interaction processes in the intention phase are characterized by several iterations between consumers and potential providers required to negotiate a preliminary agreement. Consumers initiate the negotiation processes and run them as long as at least one proposal is created (*Complex product proposal*) fitted to satisfy the requested consumer's demand.
- The interaction processes related to the contract phase, are defined by interactions necessary for creating and confirming a legally-binding contract (*Complex product contract*) based on the preliminary agreement. As with the intention phase, the interactions are initiated by consumers leading the process of creating an umbrella contract based on confirmations received from the involved providers and ordering.
- The interaction processes in the settlement phase are characterized by interactions required for the fulfillment of the legally-binding contract agreed in the previous phase. These fulfillment-related processes generate the transaction data, and together with data generated in previous phases, they build a transactional dataset (*Transaction record*). The generated transaction records serve as the basis for all subsequent processes that might occur during the follow-up phase. Moreover, these are records of the institutional history that are considered essential for building trust and reliability among the DMS as a self-organized exchange environment.

After overviewing the interaction process model at a higher level, the following sections provide more detailed descriptions of individual phase-related processes.

4.3.1. Knowledge Phase

Three main processes define the knowledge phase. As indicated in Figure 6, these are processes that enable:

- Becoming a member and joining the DMS ecosystem (all roles)
- Describing and registering offerings (provider role)
- Formulating complex product requests and matching with potential providers (consumer role)

Unlike exchange environments with a centrally provided infrastructure, the DMS ecosystem is an open and self-organized environment of different actors taking different roles. In such self-organized networks, each member is at the same time the contributor of the underlying infrastructure and participant in an exchange market built upon this infrastructure. Therefore, the first step for further members is to register and share in which role (or roles) they intend to contribute to the ecosystem.

Figure 7 shows the process of joining and becoming a member of the DMS ecosystem. The process starts with requesting access, as indicated in the upper lane. The access represents an entry-point or link that enables initial interactions with the DMS (e.g., a website or link for download of required DMS User Interface). Such access requests are processed by technology providers, as indicated in the middle lane. Technology providers are actors whose primary responsibility is to provide software/hardware resources necessary for establishing and the functioning of the ecosystem. After receiving DMS access, new actors request membership by sending membership requests, which are received by stewards. Stewards (i.e., actors responsible for the governance-related tasks) decide based on the defined rules and standards and existing entries in the institutional history. The institutional history of the DMS ecosystem represents a shared record of all registered members, their roles and relevant events, and transaction-related data. Together with rules and standards, it is an instrument for decentralized governance of a self-organized network and forms the foundation for building governance services necessary for the growth and sustainability of the DMS ecosystem. Section 4.4.2 describes these services in more detail.

The membership request can either be denied or approved. The latter creates member cards (with unique identifiers) and sends them back to the requestors making an entry in the institutional history. After receiving member cards, new actors can sign-up to the DMS, and register for different roles. Depending on the chosen role (or roles), the sign-up process might encompass further interactions with technology providers, e.g., access to other tools or services.

After joining the DMS, members taking the provider role, start the process by describing their offerings as shown in Figure 8. Offerings, structured product/service descriptions, as well as a short description of providers themselves (e.g., name, address, settlement modalities, or ratings), are registered in a distributed product catalog. The distributed product catalog follows the same principle as the DMS institutional history. It serves as an instrument for publishing offerings in a trusted network of actors rather than in a mediated product catalog as is the case with centrally orchestrated solutions.

On the other side, members taking the consumer role, start the process by formulating their demand. As shown in Figure 9, consumers are enabled to create complex product requests and match them with potential providers. The matching is done based on the data stored in the distributed product catalog. In case there are no matching results (i.e., no providers who can offer product/service combination), consumers need to modify their requests or leave the process. Otherwise, when matching results exist, a provider list is generated. The generated provider list contains all necessary data required for the addressing of identified providers and starting interaction processes related to the following intention phase.

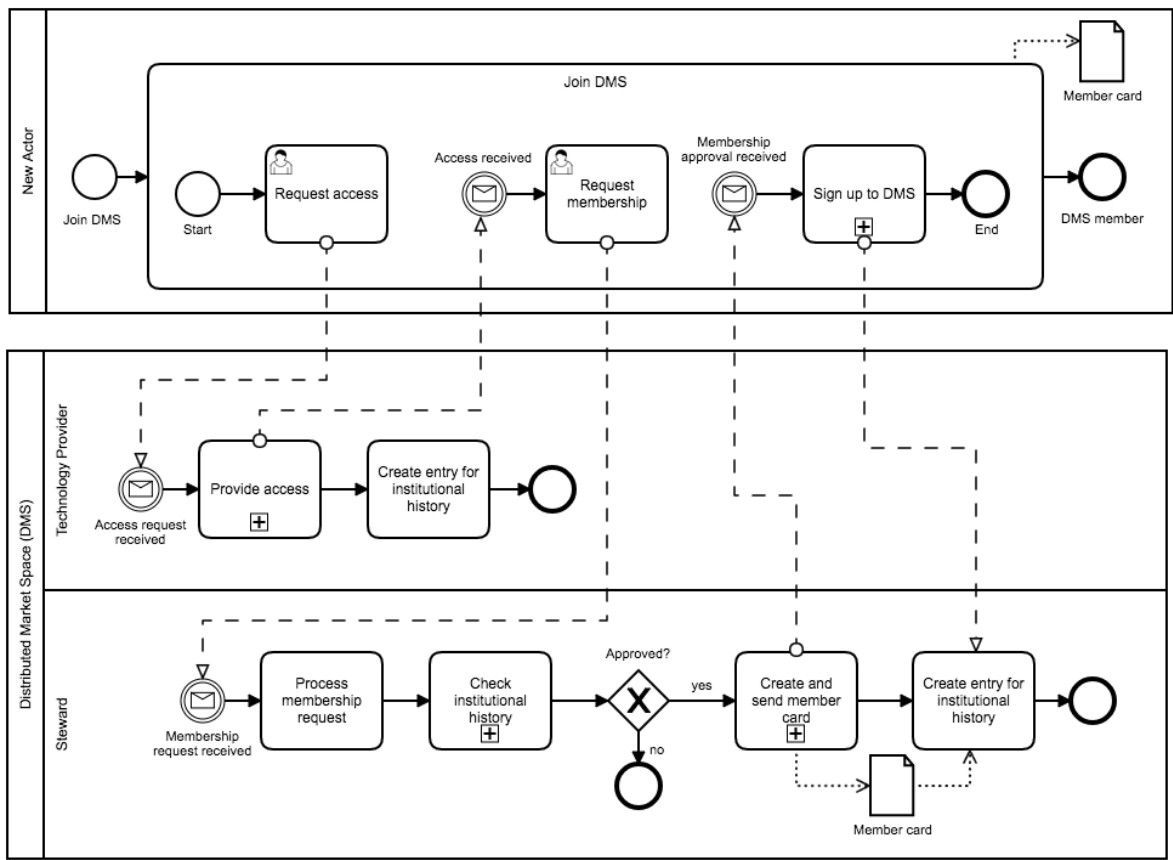

**Figure 7.** Becoming a member and joining the DMS ecosystem—a process describing a set of activities and interactions between a new actor and the enabler roles technology provider and steward, necessary for becoming a DMS member.

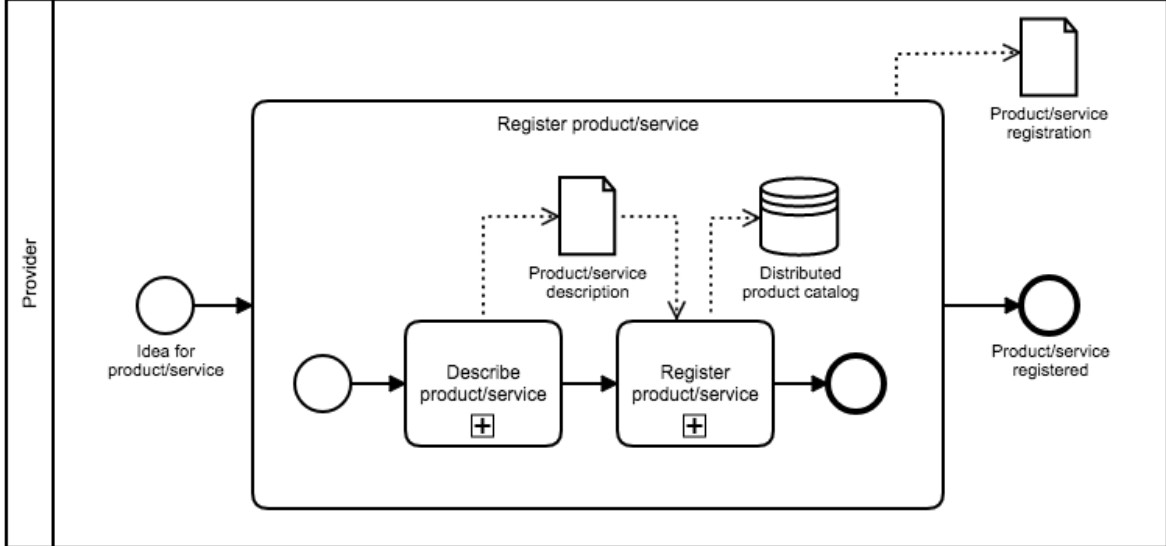

**Figure 8.** Register offerings—a process describing the provider's activities with the objective to register a product or service in the DMS.

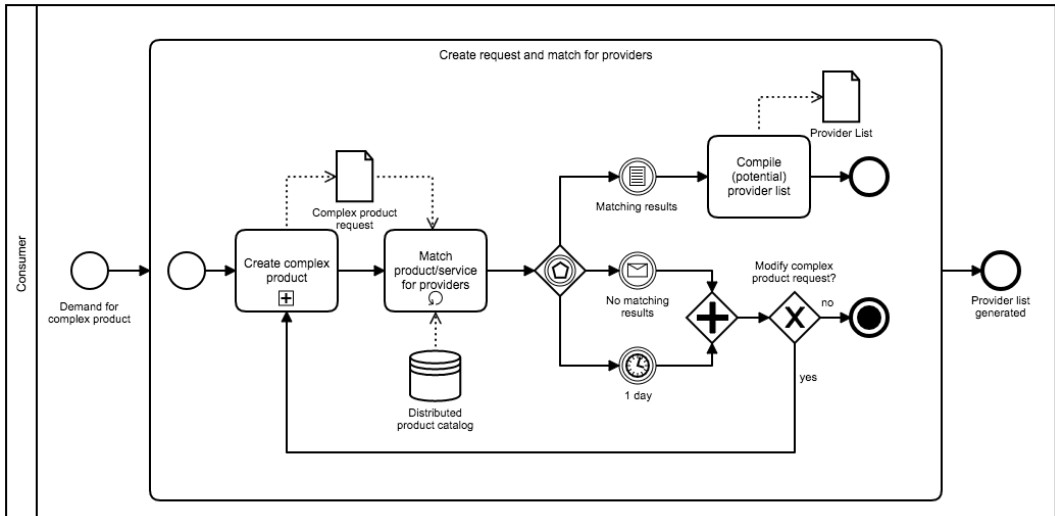

**Figure 9.** Formulating a complex product request and matching for potential providers–a process describing consumer's activities with the objective to acquire an overview of potential providers for a particularly complex request.

4.3.2. Intention Phase

After formulating demand and acquiring an overview of potential providers, consumers are enabled to initiate the intention phase. As shown in Figure 10, consumers start the process by addressing identified providers and sending them requests for offers (RfO). RfOs are requests for offers for individual parts of the complex product sent to providers. By sending RfOs, consumers signalize their intention to engage in commercial exchanges with the addressed providers. The addressed providers, on the other side, indicate their intention to provide for a particular product or service by answering RfOs and sending back concrete offerings (see Figure 10, bottom lane).

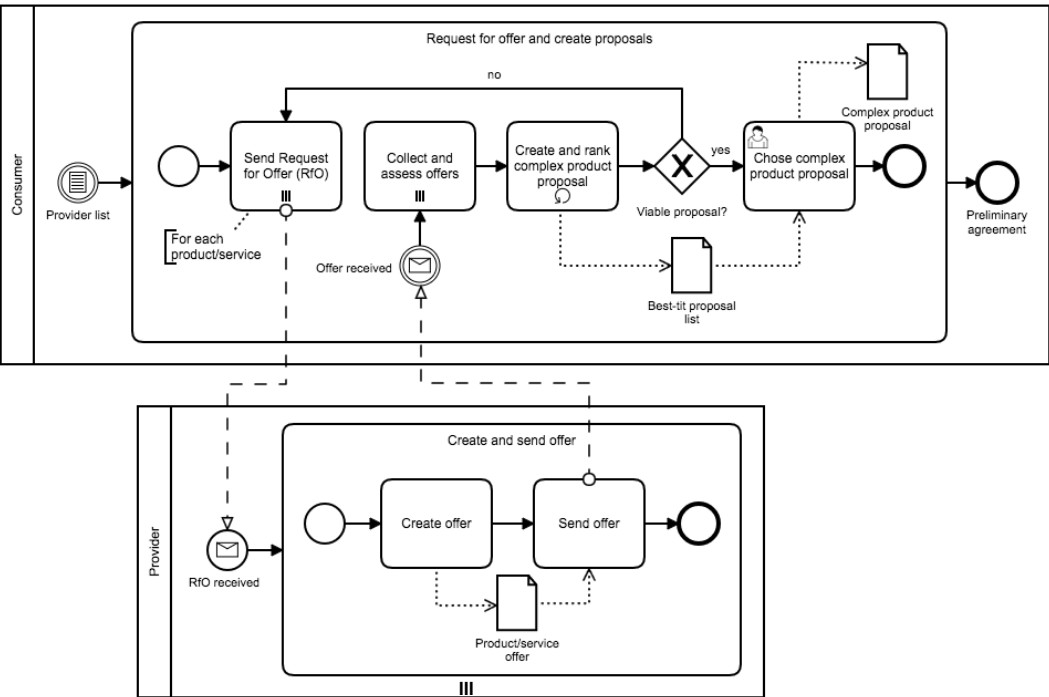

**Figure 10.** Requesting and receiving offerings, and creating viable complex product proposals—a process describing negotiation activities between consumers and providers, with the objective to achieve a preliminary agreement.

Complex products might get arbitrarily complex and include different product/service combinations. As a result, providers for each of the demanded products or services need to be addressed to obtain offerings for each of them. The related activities (i.e., requesting offerings on the consumer side and creating and sending offerings on the provider side) comprise many interactions, which might undergo multiple iterations. Such interactions might be repeated until enough offers are obtained and assessed; hence, enough viable offerings can be collected.

Collected offerings enable consumers to create complex product proposals and rank them based on their requirements and preferences. As a result, a best-fit list is created to support consumer's decision making. In case there are no viable complex proposals, consumers might go back to the beginning and start another negotiation process undergoing the activities of sending modified RfOs and waiting for providers' responses. Alternatively, and in case there are viable proposals, they might choose one to proceed. The negotiating process between involved parties ends with a preliminary agreement on a selected proposal for a particular complex product.

### 4.3.3. Contract Phase

After accepting viable complex product proposals, consumers are enabled to enter into contractual agreements with the involved providers and to conclude legally binding contracts.

Figure 11 shows the process of concluding a complex product contract. From the consumer side, a complex product contract is considered an umbrella contract since it might incorporate different arrangements for different parts. Such an umbrella contract represents a one-to-many contract situation. It requires the consumer's involvement in multiple contractual processes, one for each of the requested products or services, the aim of which being to enter into a contractual agreement with all involved providers (counterparts) for all required products or services.

The process of setting up an umbrella contract is two-staged; with stage one being considered provider-confirmed and stage two consumer-confirmed. In the provider-confirmed stage, a pending contract for each of the required product or service is created and sent to the involved providers for confirmation. Only if all addressed providers confirm all pending contracts, the second confirmation stage can start. The consumer-confirmed stage begins with placing orders for provider-confirmed (pending) contracts and receiving order confirmations. After all order confirmations are received, an umbrella contract is created to fix all confirmations.

The supporting activities for the two-stage confirmation process of a complex product contract range from creating pending contracts, sending them to providers, collecting confirmations, and finally placing orders (see the upper lane in Figure 11). On the provider side, the activities related to contracting are slightly different and considerably more straightforward. That is because the contracting process on the provider side is considered a one-to-one contract situation with an additional activity that includes the confirmation of pending contracts as the prerequisite for the final order confirmation. The setting up process ends with a legally binding complex product contract. As with all legally binding contracts, it is mandatory for all counterparts and entails all terms and conditions required for its settlement and enforcement.

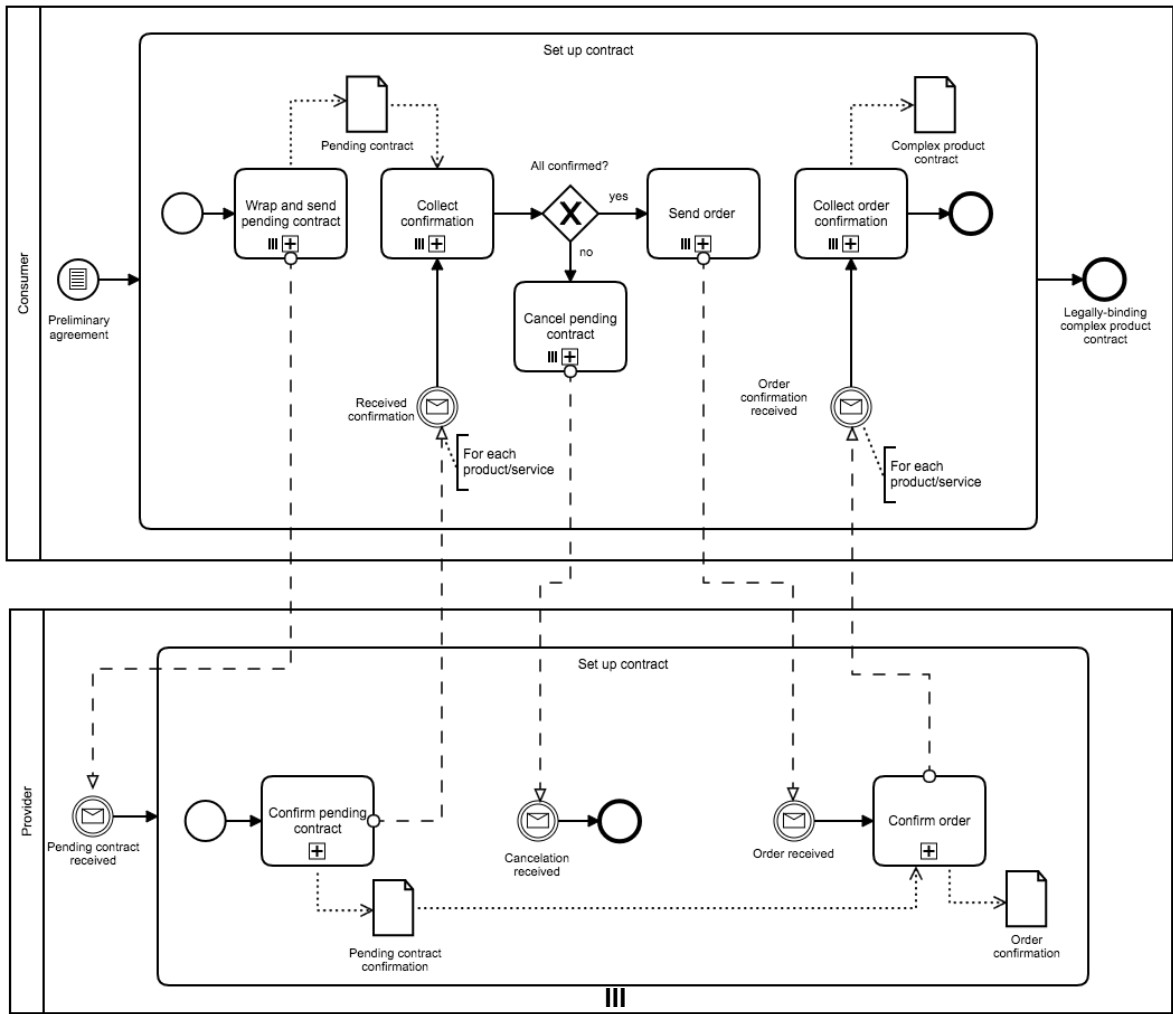

**Figure 11.** Setting up the contract—a process describing contracting activities between consumers and providers, with the objective of agreeing a legally binding complex product contract.

### 4.3.4. Settlement Phase

After setting up the contractual agreement for complex products, transactional parties are enabled to fulfill obligations resulting from this contractual agreement. These are payments on the consumer side and deliveries of ordered products and services on the provider side. Such settlement processes are well researched and understood (see [26,45,46]). The main issue with settlement processes between transaction partners who do not know each other is the lack of trust that each of them will fulfill the contractual agreement. Centralized models compensate for such trust issues via centrally organized settlement processes and by doing so, position themselves as the trusted intermediary.

Against this background and depending on the expected level of trust, we propose three different approaches regarding the settlement of complex products:

- Trusted settlement
- Trusted third-party settlement
- Trustless settlement

*A trusted settlement* assumes an adequate level of trust among transaction partners within a reference DMS. That might be the case in situations where consumers know and trust their providers, or they are involved in complex product scenarios perceived as a low-risk scenario. The settlement of standardized products and services from a low-mid price segment offered by established providers

is usually perceived less risky, as is the case with personalized products and services from the premium segment.

Figure 12 illustrates the process of fulfilling a complex product contract by applying the trusted settlement approach. On the consumer side, the process starts with payment for each product and service and sending payment notification to each of the involved providers. After the receipt of payment, the process on the provider side starts with returning the receipt of payment followed by the delivery of the order. On the consumer side, the process continues with waiting for the delivery of all orders. In case of delays or other unexpected events, consumers might ask for compensation or alternatively initiate their complaint management. In all other cases, the consumption starts after the delivery of all orders, and the whole process ends with creating a transaction record and an entry for the institutional history.

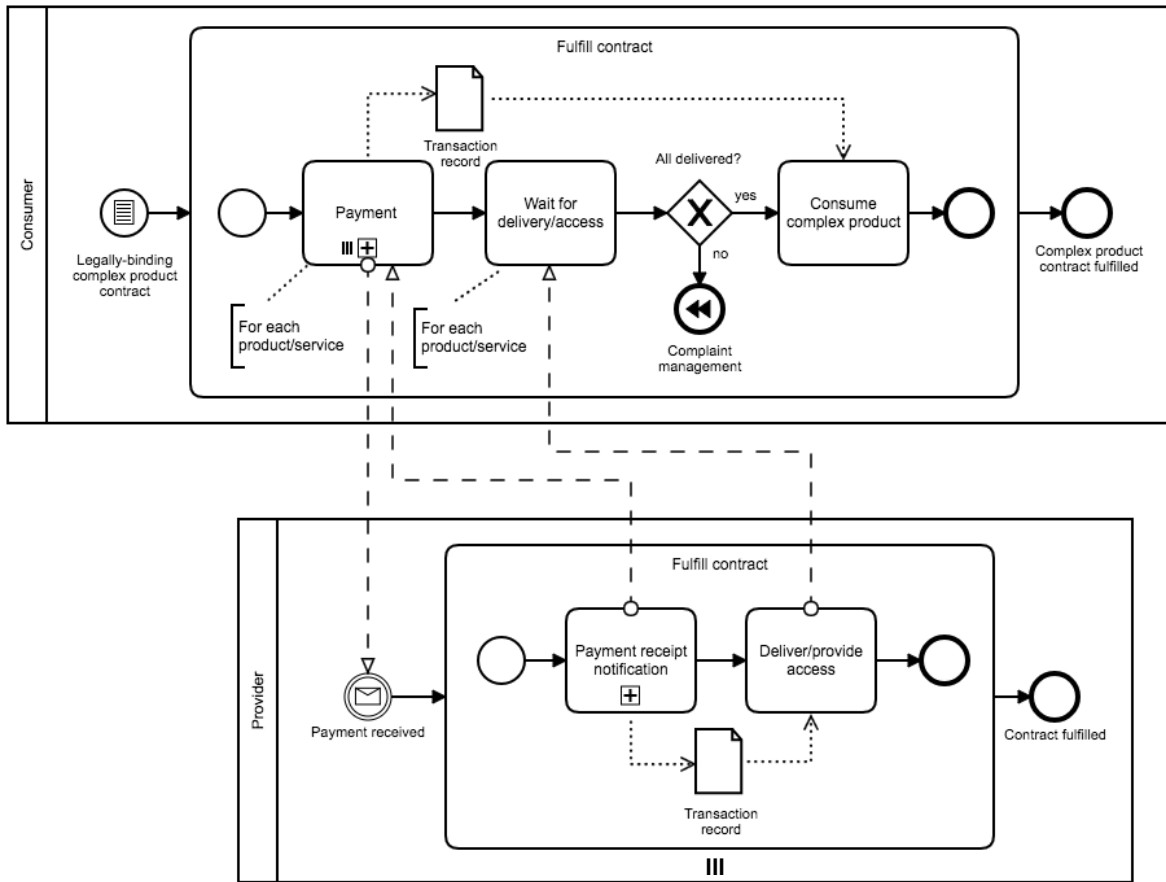

**Figure 12.** Fulfilling a complex product contract—a process describing interactions among transaction parties by applying the trusted settlement approach.

*Trusted third-party settlement* assumes engaging a third party trusted to be capable of ensuring safe contract fulfillment (i.e., payment and delivery according to the contractual agreement). In the DMS ecosystem, trusted third parties might come from the ecosystem's participants taking the mediator or expert roles as described in Section 1.3. Mediators or experts may be involved in the settlement process and support consumers and providers to fulfill obligations defined by the agreed contract. This approach is considered suitable for the settlement of product/service combinations from a higher price segment and higher complexity regarding the coordination and management of payment and delivery activities.

Figure 13 shows the process of fulfilling a complex product contract by applying the trusted third-party settlement approach. It summarizes the collaboration of involved consumers and providers with a mediator as the trusted party. The main activities of the involved mediator (see middle lane) are

to coordinate the payment process according to the agreed terms and conditions. The coordination process starts with the payment on the consumer side as the signal of the willingness to pay for the order. After receiving the payment notification from the mediator, providers deliver ordered products and services (i.e., provide access to them). Following the delivery notification from the consumer side, the mediator transfers payment to the providers and hence closes the payment transactions for the entire complex product. The settlement process ends with the sharing of the created transaction record.

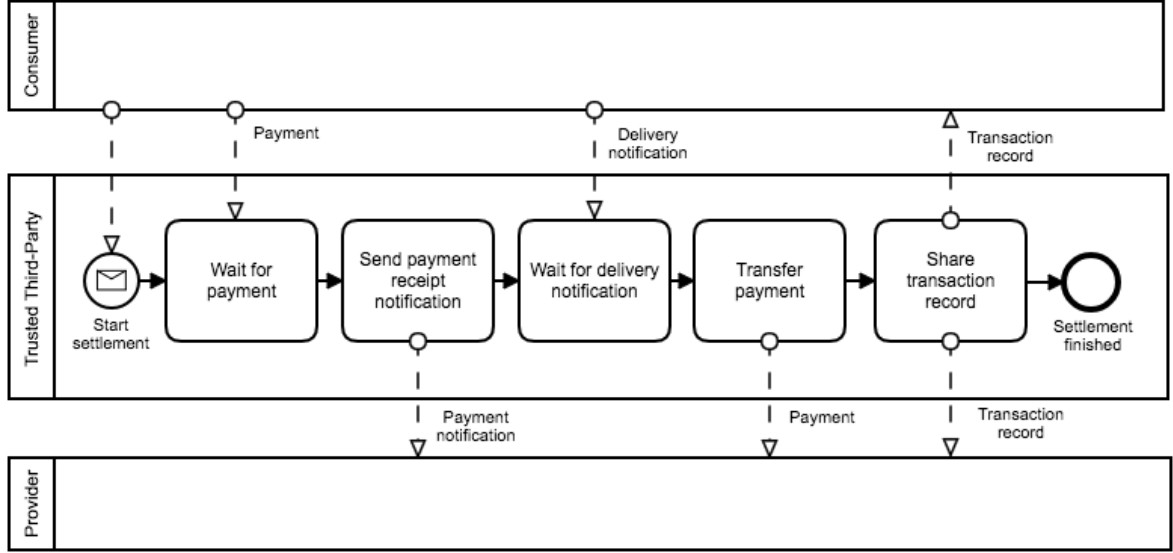

**Figure 13.** Fulfilling a complex product contract–a collaboration diagram summarizing interactions between transaction partners (consumers and providers) with a third party applying the trusted third-party settlement approach.

*The trustless settlement* approach builds on the concept of smart contracts. Smart contracts are automatable and enforceable contractual agreements, which could be implemented as immutable computer programs to run on a broader range of technology platforms, including distributed ledger platforms, e.g., Etherum [47] and Fabric [48]. Smart contracts are "trustless" since they refer to a "piece of computer code". They execute the terms of a contract and by doing so, minimize the need for trust between transaction counterparts or the need for a trusted third party [49]. By using smart contracts, the contractual clauses (i.e., terms and conditions) and interactions between transaction parties are translated into code and are transparent and self-enforced. In case any party deviates from the contractual agreement, the following actions, e.g., payment and penalty, are known and automatically enforced by the smart contract. Even though smart contracts and distributed ledger technology are still in their early stages [50,51], the trustless approach seems promising for the settlement of complex products in particular, e.g., the Internet of Things (IoT) scenarios. IoT scenarios incorporate predominately digital products and services and require many different and partly heterogeneous providers (e.g.,autonomous agents and machines) for the ordering of the complex product to be realized.

Figure 14 illustrates a simplified process of trustless contract settlement based on smart contracts. As indicated, transaction partners (consumers and providers) first need to generate the smart contract code that entails the terms and execution logic for the legally binding complex product contract. After all involved partners agree with the generated code, it is deployed in the underlying blockchain waiting to be triggered by the consumer. Once the smart contract is initiated, it executes itself. As a result of the process, a transaction record is created and shared with the involved parties.

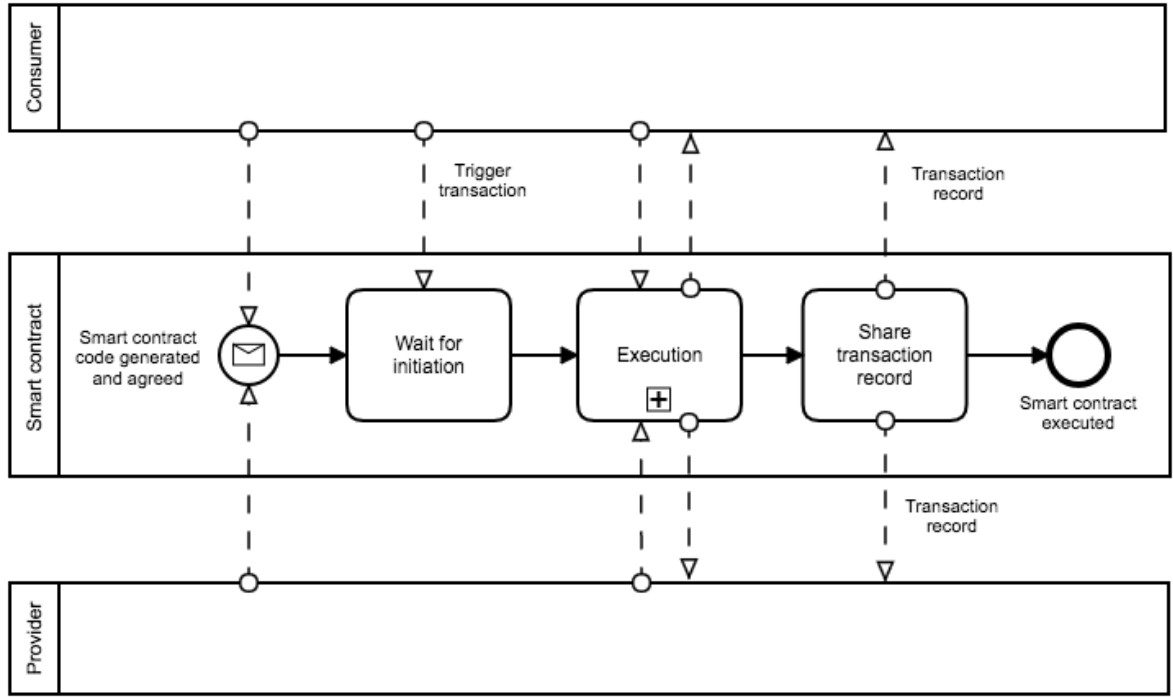

**Figure 14.** Fulfilling a complex product contract—a collaboration diagram summarizing activities related to the initiation and execution of smart contracts representing the trustless settlement approach.

4.3.5. Follow-Up Phase

The processes of the follow-up phase close the market transaction for complex products. As indicated in Figure 6 it encompasses processes to enable:

- After-sales processes
- Dispute resolution

*After-sales* processes for complex products are considered the same as for individual products and services. These are processes that enable transaction partners to review settled transactions to extend their relationship by offering additional activities such as, e.g., customer support, as well as to handle return and refund [26,46]. Reviewing settled transactions primarily related to the involved parties providing reviewers (two-side reviewers). Moreover, the reviewing process might entail the sharing of experiences with other ecosystem participants as well as the sharing of transactional data. Based on that, reviews are used to support decision making considering market transactions, but also as an indication of the ecosystem's ability to support market transactions. For a detailed description of the afore-mentioned after-sales processes, see [25].

The *dispute resolution* process is required to enable DMS participants to settle potential disputes. A dispute is considered a form of a conflict in which one party makes a claim (called filer), and the other party rejects that claim (called respondent) [52]. In recent years, diverse Online Dispute Resolution (ODR [53]) approaches were developed following, in general, the same process. Accordingly, an online dispute resolution process must allow disputants to choose an adjudicator, i.e., a mediator, who is capable of resolving such issues. The mediator, on the other hand, is required to be as neutral as possible and to contribute to the dispute resolution in an efficient and swift manner.

Following these recommendations, Figure 15 presents an exemplary process for dispute resolution within a DMS ecosystem. It illustrates relevant interactions between disputants (a filer and respondent) and a mediator who intervenes to resolve the dispute in place. In disputes related to complex products, the filer is normally the consumer, and the respondent is usually the provider. The mediator is a DMS participant taking the role of an expert or mediator (cf. Section 4.2). The dispute resolution starts

with the recognition of a dispute and the willingness of the filer and the respondent to appoint a mediator. After choosing a trusted mediator, the actual resolution process starts with the request for mediation by the involved parties. After accepting the mediation mandate, the mediator organizes hearing sessions to collect statements pertaining to the dispute by all parties involved, followed by the process of identifying and negotiating viable options. As a result, a binding resolution agreement is proposed by the mediator and in the best case agreed by disputants. The process ends with the wrapping-up of mediation records and their sharing with disputants.

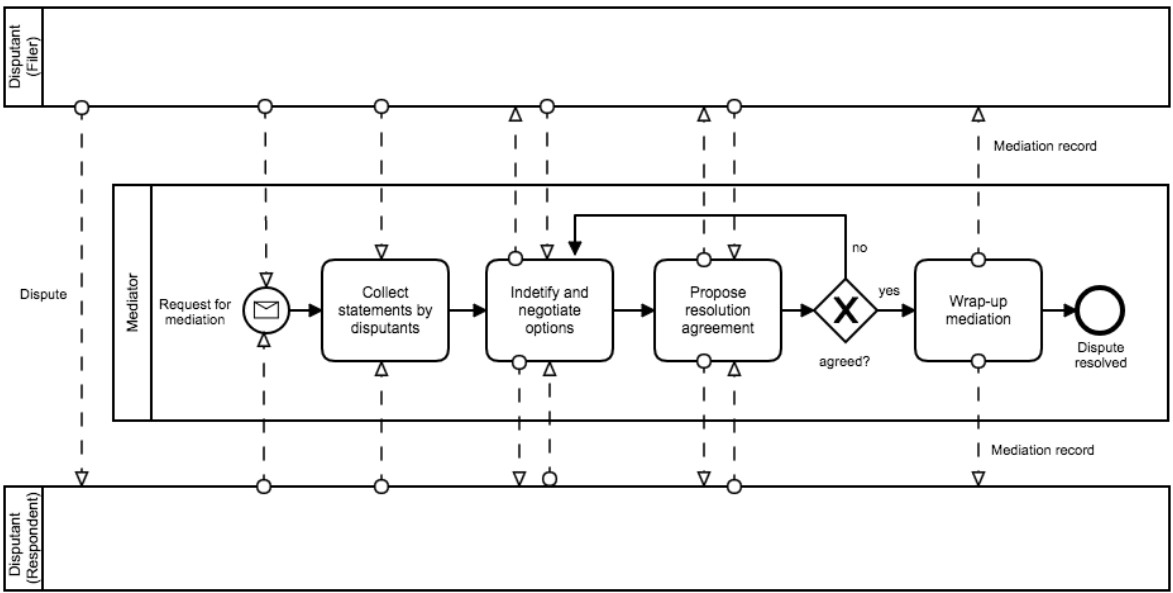

**Figure 15.** Resolving a dispute–a collaboration diagram summarizing interactions between involved parties (disputants and a mediator) necessary for achieving a resolution agreement.

In concluding, it can be noted that after-sales processes are predominately related to the follow-up phase. However, they also might take place as a parallel process in other phases, as well as intersect several phases. In particular, this is the case with sharing transactional data, which can be involved in different activities and as part of various process steps. The next section discusses the abovementioned processes in the context of services considered necessary for their implementation in a reference DMS.

### 4.4. Service View

The service view defines services that a reference DMS must provide in order to support the ecosystem's organizational structure and its core interactions. Identified services build a *Service Stack* required to implement the previously introduced ecosystem model and the related interaction process model.

Figure 16 illustrates the proposed *Service Stack*. It summarizes the identified services, grouping them according to their functions or their affiliation to a specific aspect of a reference DMS. It comprises three groups of services:

- Foundational Services
- Governance Services
- Specialized Services

As indicated in Figure 16, service groups are linked together in a hierarchical order (i.e., a stack), whereby the foundational services form the base of the DMS service stack. Based on that, governance services provide a self-governance framework that sets the desired behavior for the DMS ecosystem to which specialized services must adhere. This ensures that specialized services follow the rules

and norms that are implemented and monitored by governance services. Together, these services contribute to the ecosystem's capability to develop, sustain and provide for its health and vitality.

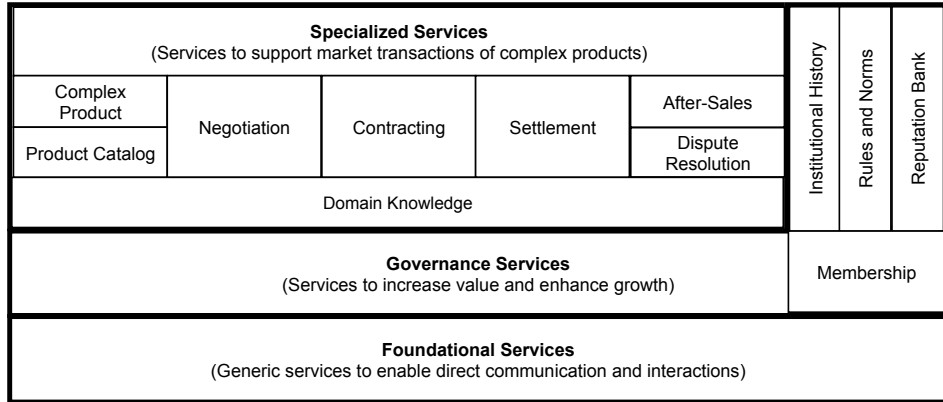

**Figure 16.** Service Stack of a reference DMS.

### 4.4.1. Foundational Services

Foundational services support the establishment and functioning of the DMS ecosystem by enabling direct communication and interaction between its participants. As a self-organized and decentralized environment, a reference DMS builds upon a network of participants who are equal in their rights and responsibilities. The main responsibilities of such networks (also referred to as peer-to-peer networks) are

- to enable discovery, locating, routing and messaging among the ecosystem's participants, and
- to assure the security and reliability of the ecosystem's communication infrastructure.

*Discovery services* enable finding other participants within the ecosystem. *Locating and routing services* optimize the path of a message traveling from one DMS participant to another. *Messaging services*, on the other hand, are critical to support direct interactions between participants. They enable addressing participants of interest and sending them defined messages.

*Security and reliability services* address two further inherent challenges for the DMS as a decentralized, networked environment. While a security service is required to ensure that only authorized participants (i.e., DMS members) should have access to information provided by other participants, a reliability service is expected to guarantee a reliable behavior within the ecosystem as a whole.

Even though considered generic, the foundational services might differ in their characteristics and performance. It is subject to the concrete implementation of the underlying P2P overlay and applied mechanisms and algorithms. This work uses CHORD [54] as the mechanism, as will be demonstrated in the case study presented in Section 6.

### 4.4.2. Governance Services

Governance services constitute the governance structure of a reference DMS as a self-governed environment with no central instance of the control. They increase value and growth and help protect the ecosystem from misconduct and outright fraud. Following the design principles for self-governed communities proposed by Ostrom [55] and the peer production approach by Benkler [56], the following services outline the self-governance structure of a reference DMS:

- Membership
- Monitoring
- Institutional History
- Reputation Bank

The *membership service* is necessary for implementing processes essential to forming the DMS ecosystem. This includes processes related to joining (and leaving) the ecosystem as well as to registering for different roles, as presented in Section 4.3.1. Moreover, the membership service is to enable transparency and clarity about who is performing, which activities in which roles and how they align to ensure that participants follow the ecosystem's code of conduct defined by rules and norms.

The *monitoring service* is required to help regulate behavior among the DMS ecosystem, which is described by rules and norms. The rules are considered explicit rules that define the terms of participation or service, and norms refer to ways of behavior that all participants are expected to align with. The monitoring service enables the scanning of "rule-breakers" and supports their enforcement. Besides, it integrates tools and services to facilitate activities of the steward role (cf. Section 4.2). These create are for example an easily accessible and well-structured documentation of the code of conduct, communication on relevant issues and events, as well as open records in order to explain their decisions and make them transparent for the rest of the ecosystem.

The *institutional history service* is necessary for the logging of events relevant to the performance of the DMS ecosystem. It implements a distributed record of all registered members, their roles and related activities, which generate transactional data. As with many self-governed ecosystems (e.g., Wikipedia), institutional history is considered a primary instrument to ensure the ecosystem's internal transparency. Institutional transparency is seen as essential to help members using existing and developing new resources [3]. Institutional transparency fosters trust among participants since trust is not a static concept and grows over time as a result of experiences and interactions within the ecosystem.

The *reputation bank service* is required for the creation and maintenance of a reputation bank that holds records about the ecosystem's members regarding their reliability, solvency, and worthiness. It implements a distributed record that, on the one hand, captures two-sided reviews about settled transactions as an indication of reliability and solvency. On the other, it may keep records about social value (i.e., a subjective form of value) members built up due to their excellent behavior and contributions to the community. The rationale behind this concept coined as "social currency" [57] is to reward a desired behavior that countributes to the assessment of the member's reputation. For example, members taking an expert or mediator role might collect credits by providing excellent services to other members, or stewards can receive credits for their engagement and contribution to the good of the ecosystem.

4.4.3. Specialized Services

Specialized services facilitate market exchange within a reference DMS by supporting processes related to market transactions of complex products, and thus implementing the interaction process model presented in Section 4.3. As indicated in Figure 16, eight services are necessary for implementing each of the five phases of the process model:

- Complex Product
- Catalog
- Negotiation
- Contracting
- Settlement
- After-Sales
- Dispute Resolution
- Domain Knowledge

The *complex product service* supports consumers in formulating their demand and acquiring an overview of potential providers for a particular complex request. It implements different process steps as illustrated in Figure 9.

The *catalog service* enables the creation and maintenance of the product catalog of products and services offered over the DMS. It assists providers in describing and registering their offerings, hence implements the process illustrated in Figure 8.

The *negotiation service* implements interaction processes related to the intention phase. It supports negotiation processes between consumers and providers, helping them to achieve a preliminary agreement for a specific complex product, as illustrated in Figure 10.

The *contracting service* enables transaction partners, creating legally binding contracts for complex products. The contracting service implements the process of creating umbrella contracts by supporting the two-stage contract confirmation process, as illustrated in Figure 11.

The *settlement service* supports the transaction partners in fulfilling contractual agreements regarding payment and delivery. The settlement service implements the different settlement approaches, namely, trusted settlement (see Figure 12), trusted third-party settlement (see Figure 13) and trustless settlement (see Figure 14).

The *after-sales service* implements processes related to the coordination and management of return and refund, reviews, and customer support. It also includes supporting transaction partners in reviewing settled transactions by giving two-sided reviews and sharing their experiences with other participants to inform their decision making (cf. Section 4.3.5).

The *dispute resolution service* implements the dispute resolution between DMS participants (see Figure 15). It facilitates achieving resolution agreements among parties involved but it also integrates other participants such as mediators or experts who might intervene as an adjudicator or an enforcer of such agreements.

The *domain knowledge service* enables the sharing of knowledge relevant for the market transactions in a particular domain (i.e., domain knowledge). Domain knowledge encompasses standardized ontologies and vocabularies, and other domain-related terms and conditions. Such domain knowledge might come from different sources, but it needs to be published by DMS members in order to be used by other services from the DMS service stack (e.g., product catalog and complex product services). That is particularly the case with members taking on the expert or knowledge provider role who are responsible for providing reliable domain-knowledge (cf. Section 4.2).

*4.5. Infrastructure View*

The Infrastructure view outlines the technical infrastructure of a reference DMS necessary for the implementation of the previously defined service stack (cf. Section 4.4). The technical infrastructure hence represents an underlying information system that implements the foundational, governance, and specialized services, and thus supports a reference DMS on the operational level.

An information system required to support a reference DMS as a self-organized and strictly decentralized market-oriented environment needs to follow the same design principle since the system design should primarily relate to its function or purpose. According to this "form follows function" principle, the underlying information system has to employ distributed resources in order to implement the DMS service stack in a decentralized manner. Therefore, the primary concern of the required system is to support actors not only as market participants (taking part in a decentralized, i.e., peer-to-peer market and its mechanism) but also as the constitutive parts, which actively contribute to these mechanisms.

In our early work [8,9], we introduced the Architecture for Distributed Market Spaces. It was purposely designed and developed to serve as a possible implementation of the infrastructure view for a reference DMS. In the following, we provide a high-level overview and brief description of the proposed architecture, as depicted in Figure 17.

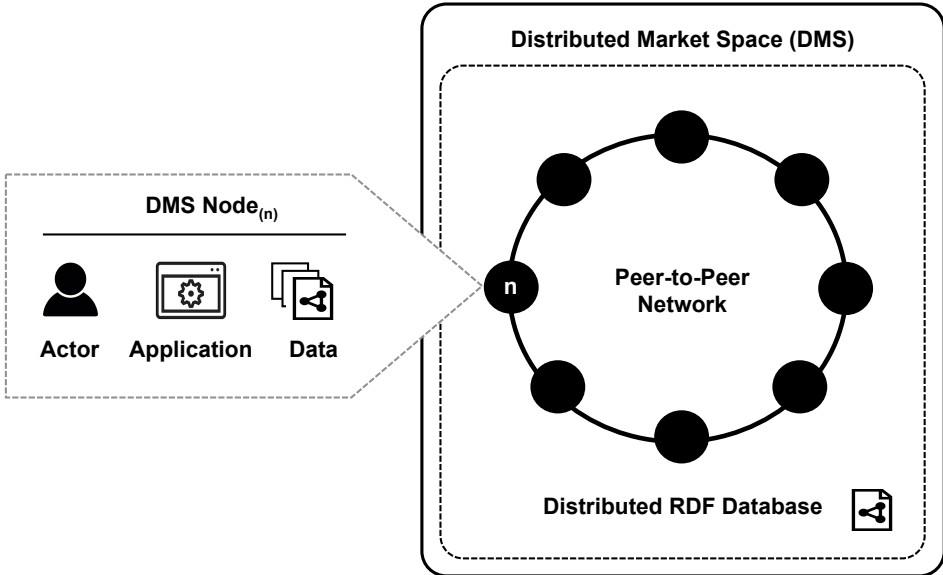

**Figure 17.** A high-level overview of the architecture for distributed market spaces ([8,9]).

An *actor* can be everybody or everything connected to the Internet defined by the intention to participate in the ecosystem. Actors join the ecosystem by connecting to its underlying network, which is organized as a structured *peer-to-peer network*. The primary responsibility of the underlying *peer-to-peer network* is to enable direct communication and interactions among actors and thus to implement foundational services (cf. Section 4.4.1). As a result of these direct connections, each actor makes a two-fold contribution to the DMS. Firstly, actors are constitutive parts of the system. They constitute the underlying *Peer-to-Peer Network* and by doing so, facilitate the ecosystem to build on top of this network. Secondly, actors are users of the system, and they are taking different roles. As such, they contribute to the ecosystem's organizational structure, but at the same time, they satisfy their motivation for participation within the ecosystem.

The *DMS Node* is the representation of an actor within the DMS. It implements the functionality of governance (cf. Section 4.4.2) and specialized services (cf. Section 4.4.3). These are provided by the user application, which runs on each DMS node. The user application connects actors to the ecosystem and supports the actor's activities related to different ecosystem roles, as well as to different steps of market transactions for complex products.

The *distributed RDF store* represents an organized collection of information necessary for the functioning of the DMS. This is, on the other hand, information relevant for the upholding of the ecosystem's organizational structure. On the other hand, this is domain-related information necessary for the market transactions in that particular domain. This information is encoded using RDF (Resource Definition Framework [58]) and stored on connected *DMS Nodes* by using the operations of the underlying *Peer-to-Peer Network*. As a result, each *DMS Node* provides resources for the distributed storage and hence stores a fragment of the global data storage. This provides inherent scalability, as an increasing number of *DMS Nodes* automatically provide more resources in the underlying network.

The prototypical implementation of the architecture for distributed market spaces [8] is used for the instantiation of a reference DMS as will be demonstrated in the case study in Section 5.

*4.6. Life Stages of Distributed Market Spaces*

This section focuses on the third dimension of the reference model for distributed market spaces. The *Life Stages Model* represents that dimension and covers three stages a DMS might undergo during its lifecycle, i.e. *Design*, *Ignition*, and *Maturity*. It follows the principle of separation of concerns, where each of the life stages has different concerns and, therefore, different priorities, which in turn necessitate different activities in order to reach the threshold for the next stage.

Figure 18 presents the proposed life stages model for distributed market spaces and summarizes the concerns of each of the three stages, as described in the following. In addition, Table 2, provides an overview of activities and tasks related to each of these stages, linking them to the ecosystem, interaction, service, and infrastructure view.

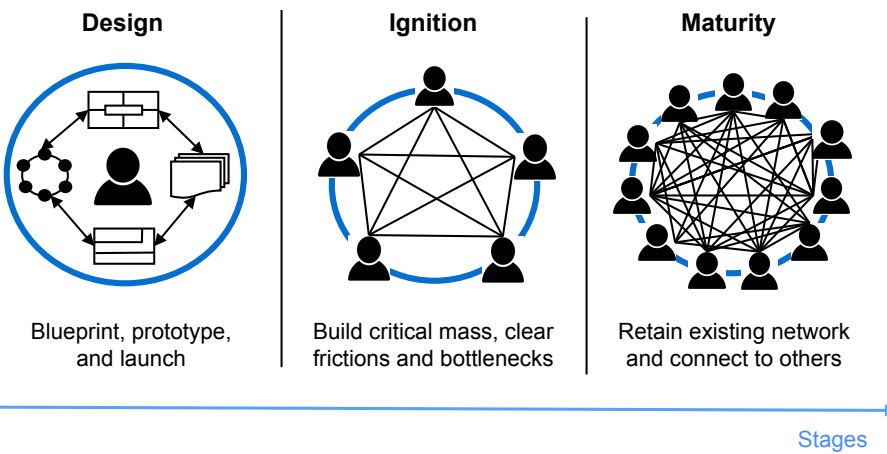

**Figure 18.** Life stages model of distributed market spaces.

**Design Stage**

The priority of the design stage is to proof the design hypothesis via a prototypical implementation for an application context. Therefore, *the primary concern is to blueprint, prototype, and launch an instance of the DMS that considers the contextual requirements of the particular application*. The blueprint is a conceptualization of a DMS instance that considers different modeling view on the one hand and integrates the application context on the other hand. As a result, a DMS blueprint comprises four extracted models, as indicated in Figure 18. Together the extracted ecosystem model and its accompanied process, service, and infrastructure models shape the conceptual structure for the DMS instantiation in that particular application context. Some of the significant *challenges* in this stage might be caused by a lack of understanding of the application context. However, modeling decisions may be based on wrong assumptions about the application context. The considerations from practitioners reference models analyzed in Section 3 suggest that the best way to start designing an online exchange environment is to focus on one single value proposition that is supported by one core interaction.

**Ignition Stage**

The priority of the ignition stage is to build the critical mass of participants in order to establish the DMS as a market-oriented environment. This network is, on the one hand, a network of participants (shaper and enabler roles), and on the other hand, a network which provides the necessary infrastructure since each of the DMS participants has a dual role. Given that, *the primary concern is to build a critical mass and clear friction and bottlenecks* in order to ensure its functioning on the operational level. One of the main *challenges* of this stage can be to "miss momentum" after the launch, i.e. to miss the momentum for gaining the critical mass of participants (consumers and providers) and consequently establishing positive network effects. Various strategies exist to cope with this issue; they propose different approaches to attract new participants in order to make the market-oriented environment more valuable for further participants (e.g., strategies by [3]). Another *challenge* related to this stage may be to miss establishing an adequate level of trust within the ecosystem. Therefore, the emphasis in this stage needs to be on the implementation of services which contribute to the ecosystem's ability to guard against misbehavior and outright fraud (i.e., governance services, cf. Section 4.4).

**Maturity Stage**

The priority of the maturity stage is to preserve the ecosystem and ensure the ecosystem's resistance and health. Resilience and health refer to the ecosystem's capability to face and survive

disruptions and to continue to be productive in creating value for all participants [59]. Consequently, *the primary concern is to retain the existing network* as the source of value creation *and connect to other networks* to gain new value sources for development and growth. Once this process has been triggered and a critical mass reached, the main *challenges* of the maturity stage can be to protect the achieved position. Defending its position includes responding to competitive threats, which may come from "within" (e.g., envelopment attack) or "outside" from other ecosystems (e.g., conglomerate or intermodal attacks) [60]). Envelopment attacks usually come from participants who established themselves as influencers. For example, a provider who uses his influencer position to develop interconnected ecosystems and take away the network or at least part of it. Outside threats, on the other hand, point to the sharing of stable networks and the building of "conglomerate ecosystems". Alternatively, they point to a more radical approach aiming to eliminate the competitive ecosystems by taking them over, as is the case with intermodal attacks. However, mature ecosystems have different possibilities to react and defend such attacks. The business ecosystem health concept [61] and the "5E" approach [6] provide an overview of strategies that are helpful in dealing with such challenges and preserve the ecosystem within the maturity stage.

**Table 2.** Life stages of distributed market spaces, and their concerns and activities linked to the dimensions of the views.

|  | Design Stage | Ignition Stage | Maturity Stage |
| --- | --- | --- | --- |
| Concern | Blueprint, prototype, and launch. | Build critical mass, clear frictions, and bottlenecks. | Retain existing network and connect to others. |
| **Ecosystem View** | Define value proposition and outline ecosystem structure. | Review participants' experiences, eliminate frictions. | Alter and extend ecosystem structure. |
| **Interaction View** | Define core interactions and related processes. | Improve core interactions and clear bottlenecks. | Review existing and propose new core interactions. |
| **Service View** | Define services to implement core interactions. | Improve and enhance existing services. | Optimize by adding new functionalities and introduce new services. |
| **Infrastructure View** | Prototype technical infrastructure and launch. | Remedy deficiencies, improve, and scale-up. | Keep infrastructure up-to-date, expand for more service. |

## 5. Case Study: Application in the Smart City Context

After introducing our reference model for distributed market spaces in the previous section, this section illustrates its application. It demonstrates how the proposed reference model has been used for the analysis, design, and implementation of a smart city project called *wemarket* (http://www.wemarket.space). The following provides an overview of the wemarket project regarding its context, goals, and scope.

**Smart City Context**

The promise of smart city projects is mainly to facilitate everyday city life by providing useful services that can be consumed by its citizens. As services derive information from data usually gathered from either IT systems or sensors, many smart city projects focus on the role of information and communication technology (ICT) as the foundation that allows for the creation of a smart city [62]. However, as cities are complex social systems [63], they cannot be reduced to their underlying ICT infrastructure. Authors in [64] propose a more comprehensive definition viewing a smart city as "an ICT-based infrastructure and service environment that enhances a city's intelligence, quality of life, and other attributes like, e.g., environment, entrepreneurship, culture or transportation." We align

with this definition and argue that it might be extended in a way to embrace value communities and ecosystems as additional aspects of a "city's smartness."

Accordingly, we define a smart city as "a service ecosystem that enhances a city's intelligence, quality of life, and adds value to its participants by facilitating seamless consumer experiences". Seamless consumer experience is the internal and subjective response consumers have to any direct or indirect interaction with the service ecosystem. Our definition of "seamless consumer experiences" leverages the definition of "customer experiences" provided by [65,66]. It extends the scope of customer experience from interactions with single service providers towards the ecosystem as a whole.

**Goal, Scope, and Assumptions**

Based on these considerations, *the main goal of the wemarket project is to facilitate a city's ecosystem formed by participants who connect to constitute a service environment for the city*–and in this way establish a city's ecosystem that emphasizes seamless experiences, rather than being a hub of individual platforms as currently is the case with, e.g., city's platforms for mobility, transportation, or city events.

The scope of the project is limited to the design, prototyping, and launching of an instance of wemarket to support application scenarios, as will be shown in Section 5.2. The main assumptions underlying the wemarket project are the following:

- The wemarket project addresses the leisure and cultural landscapes of a city and is aimed at supporting seamless consumer experiences for so-called city essentials. City essentials combine services such as leisure, transportation, and cultural events that are usually offered in a city. However, in their nature, they are highly personalized by consumers and their contextual requirements and thus regarded as complex services (i.e., complex products).
- The wemarket project is based in Frankfurt am Main, Germany as an exemplary city. This city was chosen to demonstrate aspects relevant to the contextual requirements and constraints of city essentials (e.g., location and other contextual data).
- The wemarket project was realized by a core team. It was trained and skilled to use the proposed reference model adequately and efficiently.

*5.1. Approach and Outcomes*

The case study was conducted following the recommendations of the design stage of the proposed Life Stage Model (cf. Section 4.6). Figure 19 depicts the applied process, including the four steps in which an instance of wemarket is blueprinted, prototyped, and launched. The outcomes of each of the activities are summarized in Figure 20 and explained in the following.

**Step 1 Outlining Ecosystem**

The value proposition of the wemarket.space ecosystem is determined by the project goal and formulated as *a space for seamless consumer experiences for city essentials*.

Activities necessary to be conducted by the actors (i.e., the ecosystem's participants) are activities related to the ecosystem foundation and market transaction of city essentials. Actors who are expected to participate and accomplish these activities are inhabitants, including visitors and tourists, businesses, and other city's stakeholders such as institutions, municipalities, and the city's administration. Roles these actors might take on are the roles of consumer and provider as the shaper roles. Realistically, the enabler roles (steward, knowledge provider, and technology provider) will be taken by the core team for the duration of the design stage. This is to ensure a coordinated instantiation of the service environment in the first place, and it is subject to change within the ignition stage.

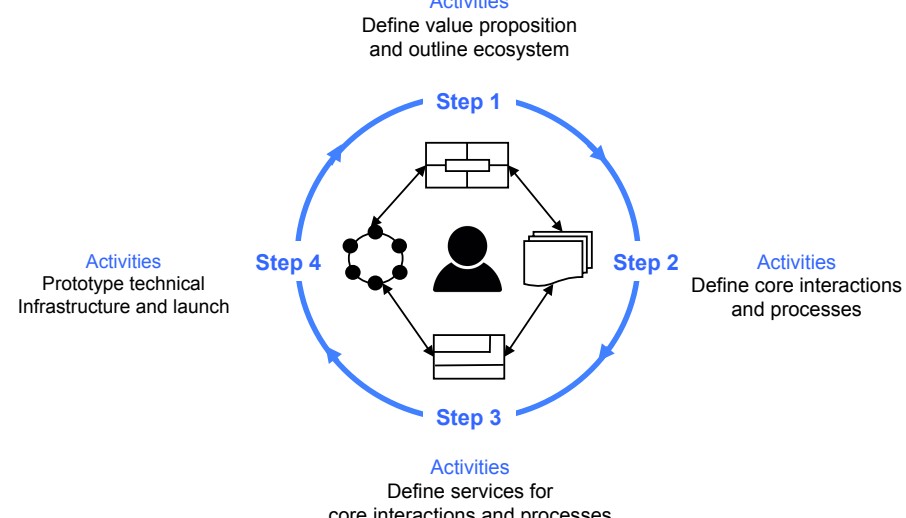

**Figure 19.** Design process showing steps for blueprinting, prototyping and launching of wemarket.space.

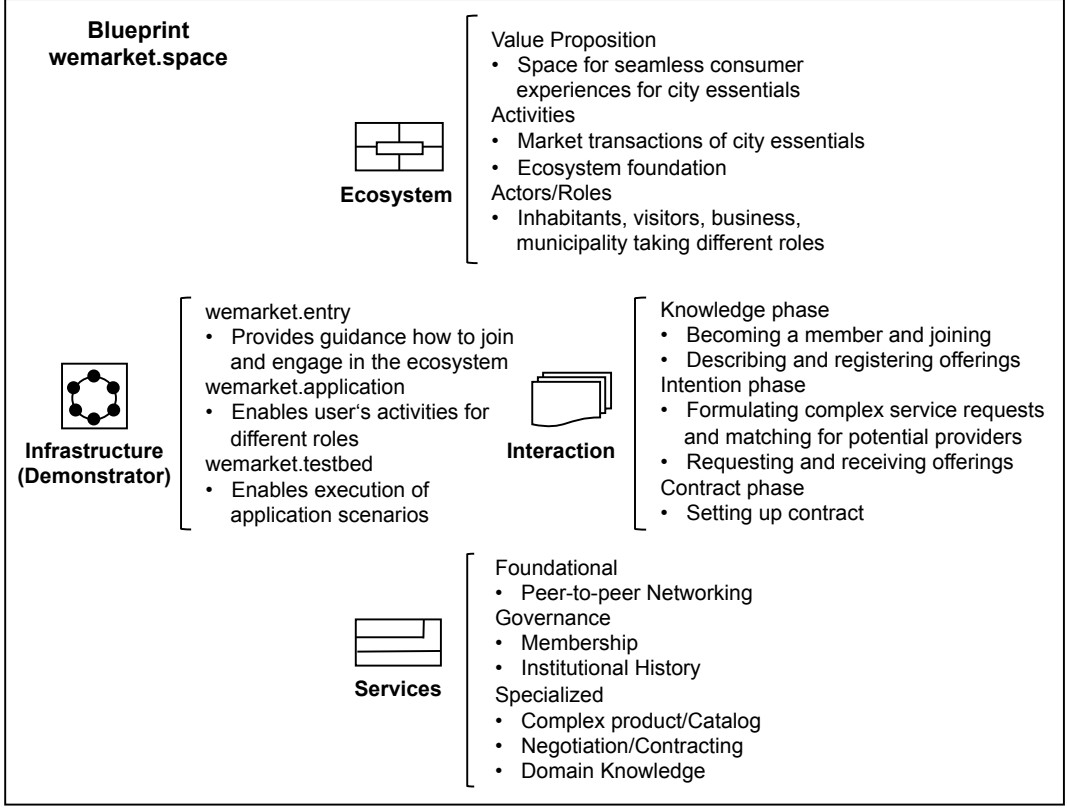

**Figure 20.** Blueprint of the wemarket.space summarizing the outcomes of the design process (cf. Figure 19).

### Step 2 Defining Core Interaction

For wemarket, three core interactions are chosen to start with. These relate to the first three phases of the interaction process model (cf. Figure 6) and hence the processes of the knowledge, intention, and contract phase. The settlement of contracts is considered as trustful settlement. It relies on direct communication with involved providers and the agreed payment and delivery modalities (e.g., cash, credit card or online payment modalities). The fully fledged settlement (supporting trustless and trusted third party settlement), and after-sales activities, forms part of the ignition stage.

**Step 3 Defining Services**

Based on the chosen core interactions, a set of services is advised. These are foundational services for peer-to-peer networking. Besides, a basic set of governance services (membership, and institutional history) is included in order to facilitate the essential self-governance mechanisms of the ecosystem. For the implementation of knowledge, intention, and contracting processes, further services have been selected: complex product, catalog, negotiation, contracting, and domain knowledge.

**Step 4 Prototyping and Launching**

The infrastructure of the wemarket has been prototyped based on the implementation proposed in [8] and launched as a demonstrator in a testbed environment. The demonstrator is realized as a Web Application (JavaScript, HTML, and CSS). It is composed of two parts *wemarket.entry* and *wemarket.application*, which have been published at the following URL http://www.wemarket. space. Thus *wemarket.entry* represents a permanently accessible node, that provides information about wemarket.space and guidelines for joining and participating. It realizes the process of becoming a member and joining the wemarket.space, and hence demonstrates the bootstrapping of the underlying peer-to-peer network. The *wemarket.application* runs on each participating node represented by a web browser instance. It enables user's activities for consumer and provider roles and currently supports human interfaces only. It supports the handling of city essentials scenarios and thus illustrates the market exchange via wemarket as the service ecosystem.

In addition, *wemarket.testbed* was implemented to enable the demonstration of different application scenarios. It includes a basic network configuration that was set up to ensure a sufficient number of members are available to accomplish the bootstrapping and thus enables new users to join. Besides, it includes a basic set of domain ontologies for the description of city essentials; these are, e.g., ontologies for domain ticketing, gastronomy, parking, and babysitting, as these domains are relevant for the demonstration scenario (see Section 5.2). Finally, the implemented testbed holds an initial catalog of providers. It was instantiated to enable the initial matching of demanded city essentials for potential providers. Provider's data sets have been generated based on real-world entries (i.e., YelloPages, http://www.gelbeseiten.de). Each of them is considered a wemarket member registered to provide for services in the domains mentioned above. The description of the wemarket.testbed and sample resources are provided at the URL http://www.wemarket.space/testbed.

*5.2. Demonstration*

After an instance of wemarket has been designed, prototyped, and launched, this section demonstrates its core interactions taking the consumer's perspective. For an exemplary application scenario, the authors recall the use case of our couple planning an evening out with friends. As explained in Section 2.1, our couple is looking for a bundle of services including tickets for a concert, the reservation of a table at an Italian restaurant, finding parking close to both locations, and engaging a well-rated babysitter to watch after their children.

The demand spans four different service domains (i.e., ticketing, gastronomy, parking, and babysitting) and has to consider contextual information regarding the couple's schedule, location, and ratings of a particular service.

To start planing, our couple first needs to become a member by visiting the wemarket entry point (http://www.wemarket.space/entry). As a new user, our couple needs to provide a few pieces of information and submit a request for membership (see Figure 21). This request is sent to the network, that is, to the ecosystem's users responsible for the processing of membership requests (i.e., steward role), and ultimately bootstrapping (see Figure 22 for interactions inside wemarket.space).

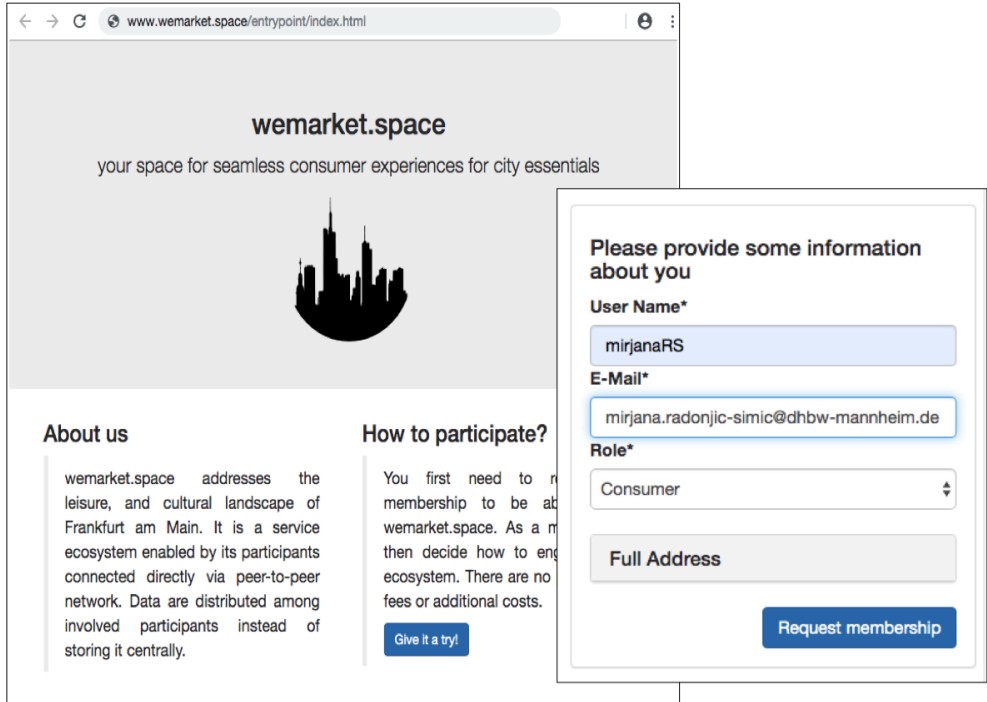

**Figure 21.** wemarket.space-new user requesting membership.

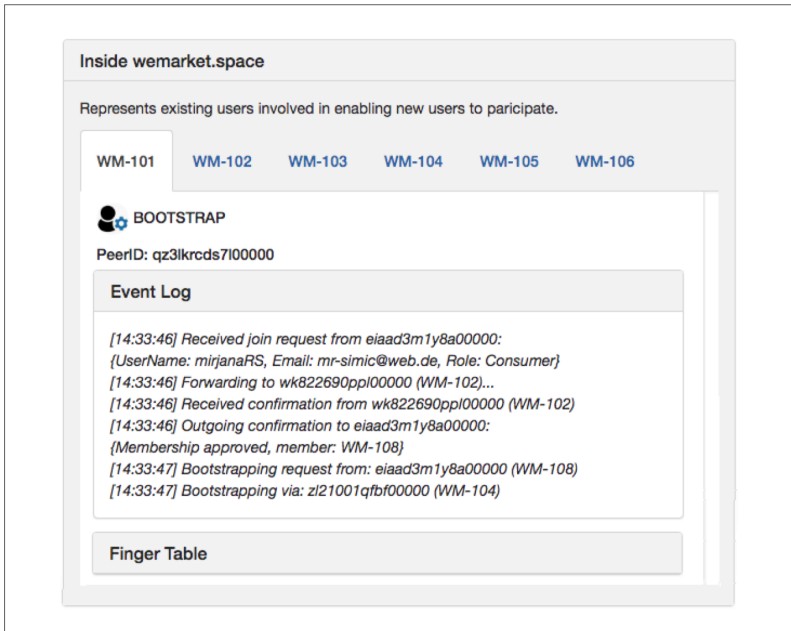

**Figure 22.** Inside wemarket.space—the processing of membership request and bootstrapping.

As a result, our couple becomes a member (and receives a membership card as a means of identification within the ecosystem), and a peerID necessary for the direct interactions among the network. This is illustrated in Figure 23. As a new user, our couple can continue with the description of the concrete demand, as shown in Figure 24. In this case, the selection of domains that describes the desired service combination. Thereby for each of the services, different fields have to be filled out (i.e., a ticket for the musical Chicago, described by name, category/genre, and price).

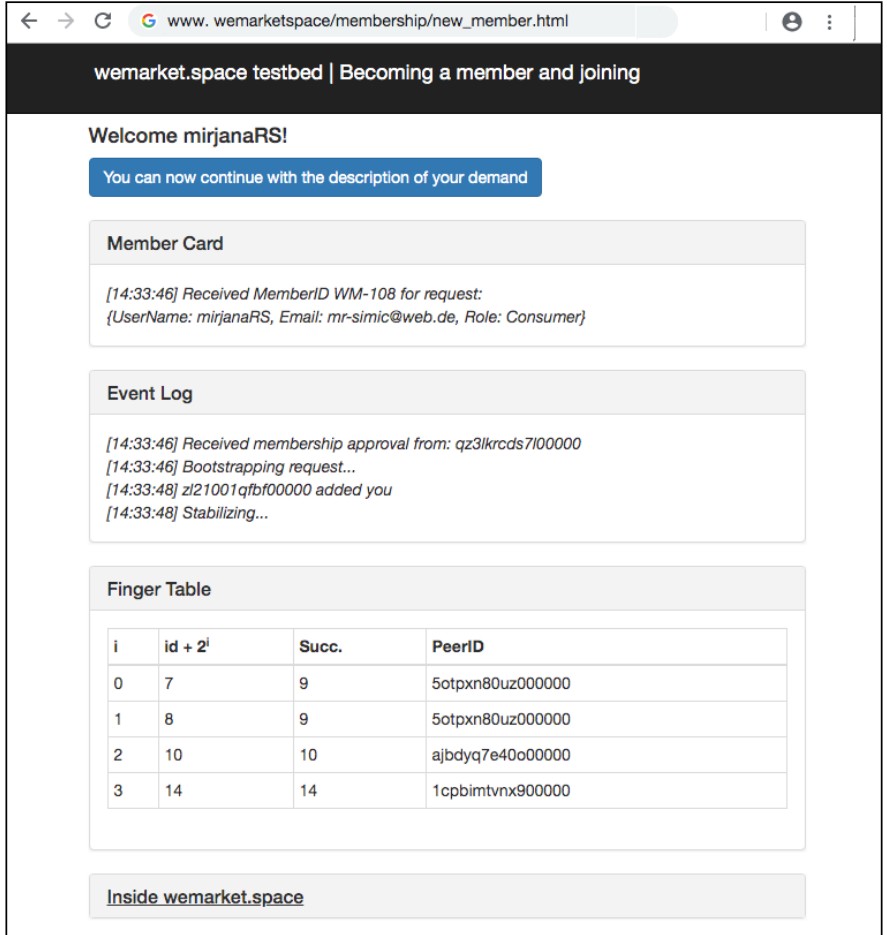

**Figure 23.** wemarket.space—new user becoming a member and joining the network.

The created request is then published within the network by clicking the button Request for Offers (RfO). This starts the matching and, thus, interactions of the consumer with potential providers for the demanded services. This is illustrated in Figure 25.

The right side of Figure 25 represents the addressed providers, which are grouped by the domains in which they offer services of interest. Incoming requests for offerings and outgoing offerings to the requesting consumer are shown. On the consumer side, all received offerings are presented in the tab (incoming). Offers, and therefore, viable proposals are created and ordered in the adjacent tab (see Figure 26). By accepting a particular proposal, the process of setting up an umbrella contract is started, and with that the realization of the two-stage contract confirmation process. After all pending contracts have been confirmed, the consumer can finish the process of creating a legally binding contract, and each involved provider then receives the order confirmation, as the legally binding agreement.

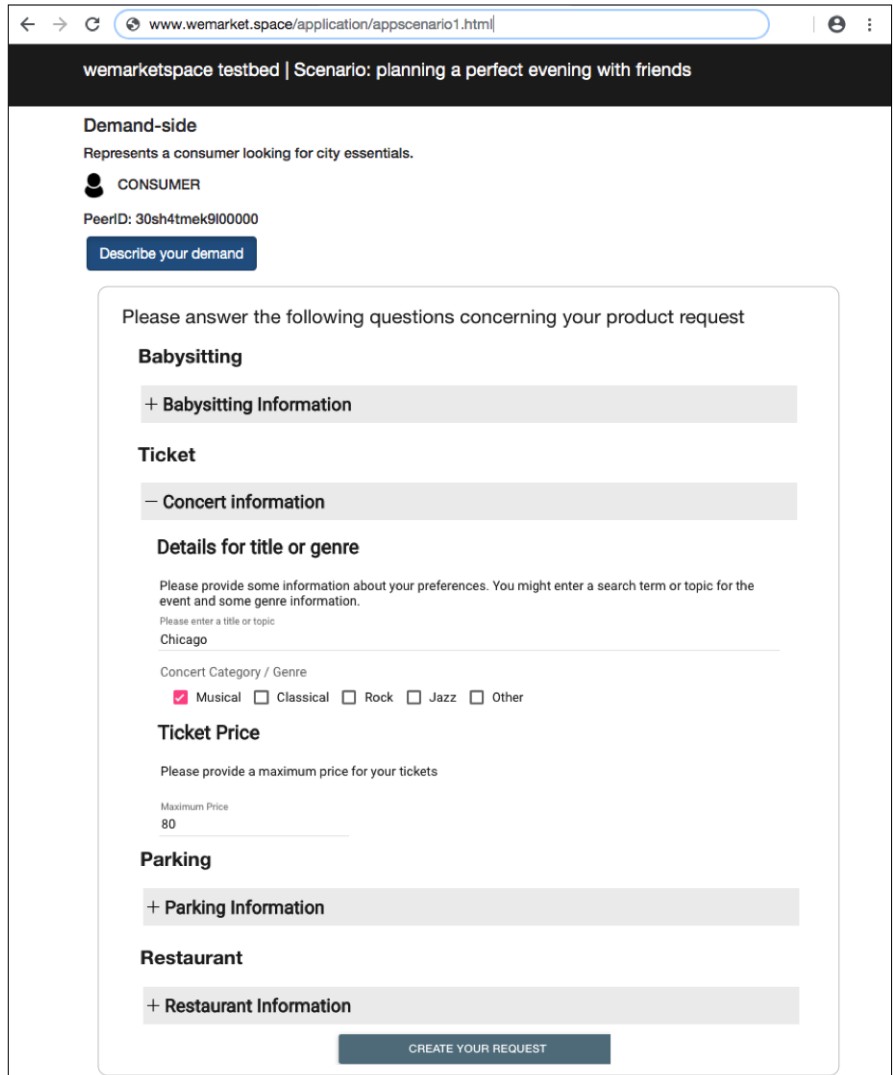

**Figure 24.** wemarket.space—description of demand, data entering.

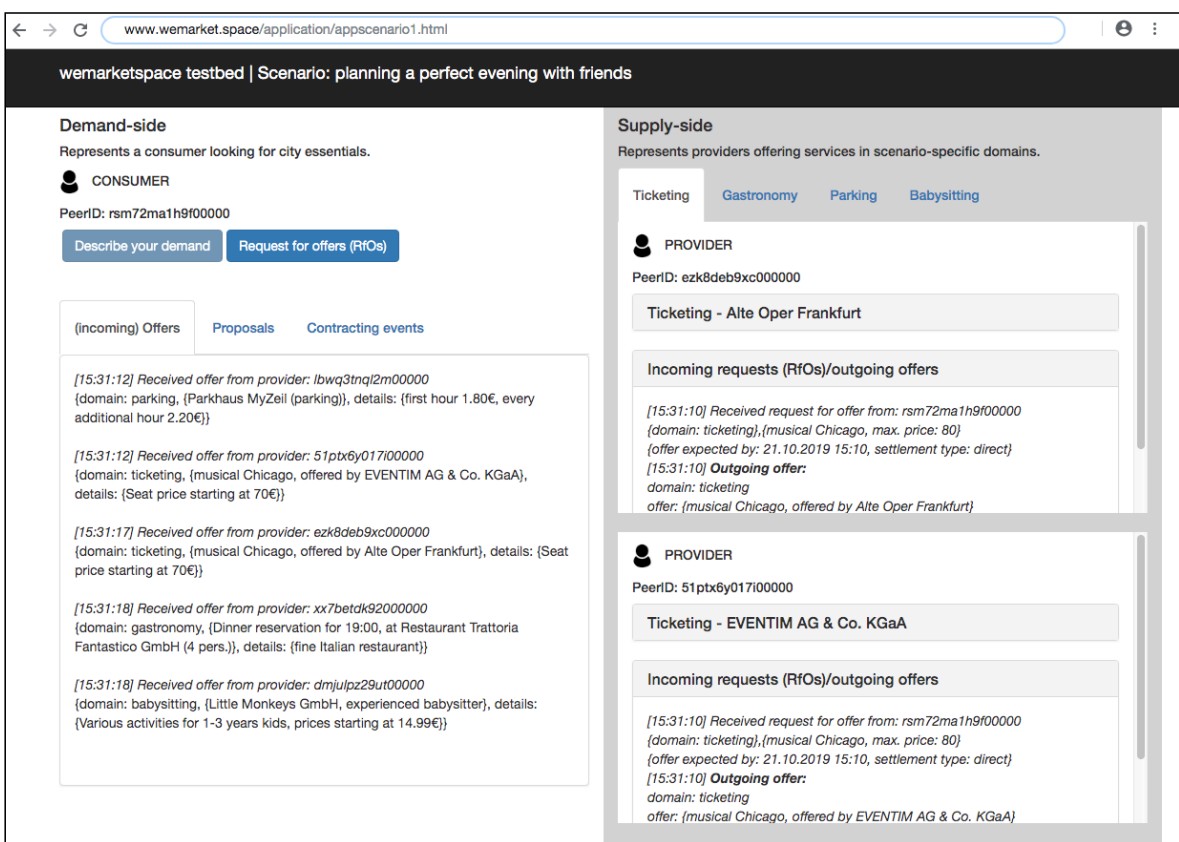

**Figure 25.** wemarket.space—matching, requesting for offers, and receiving offers.

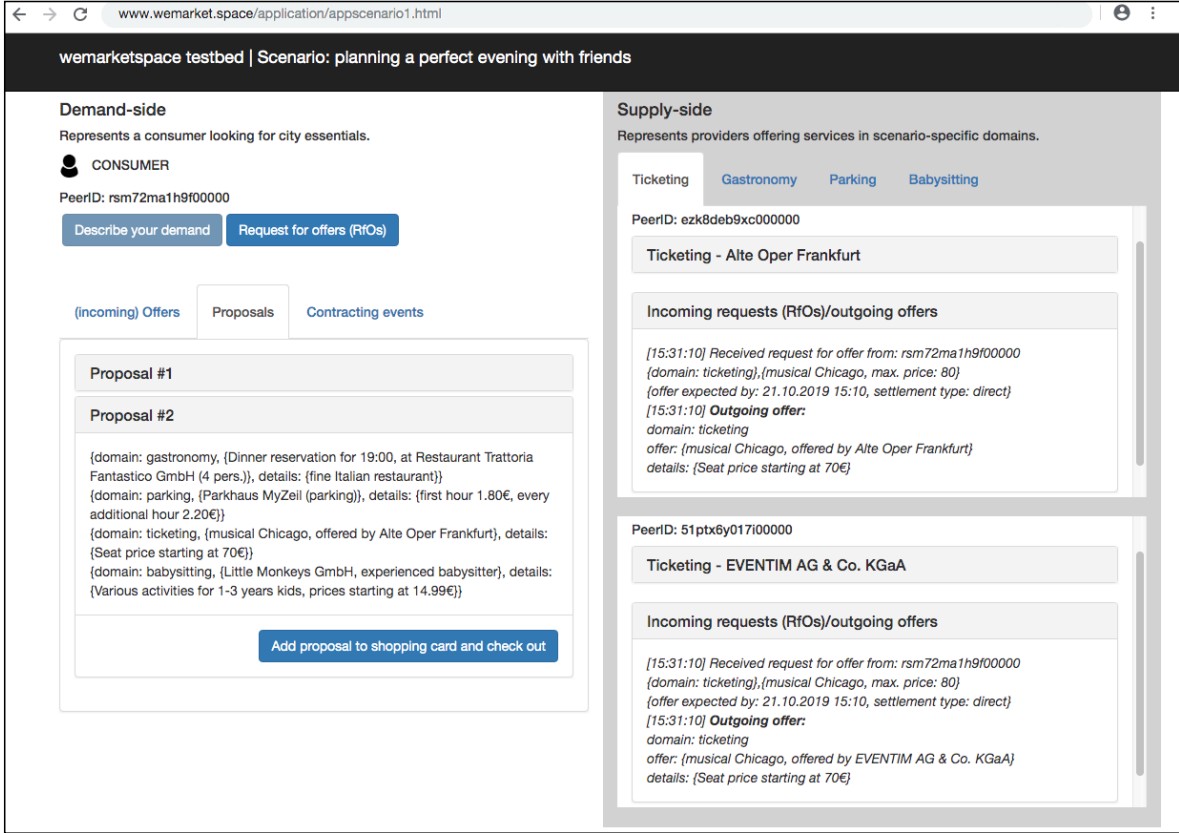

**Figure 26.** wemarket.space—creating proposals and starting contracting.

## 6. Discussion

The main goal of the case study was to demonstrate the applicability of the proposed reference model in the context of the wemarket project. The focus was on the ignition stage (cf. Section 4.6) and the process steps related to the blueprinting and prototyping of a wemarket.space instance, which demonstrates the core interactions among shaper roles (consumers and providers), thus to show how a service ecosystem can be established upon the proposed software-architecture as the implementation of the DMS service stack (cf. Section 4.4). On the one hand, this simplifies the scope of the case study to some extent. Still, on the other hand, it emphasizes design and instantiation as the foundational steps for the testing of blueprinted models (ecosystem, interactions, services, and infrastructure, cf. Figure 19 ).

The results of the conducted case study showed that the proposed reference model for distributed market spaces meets the overall objective of serving as a guidance for analysis, design, and implementation of decentralized and self-organized structures.

The smart city context, as defined in this paper, and the wemarket project represent a post-platform scenario for two reasons. Firstly, wemarket recognizes consumers and providers as the primary drivers of the economic exchange in a smart city as a service ecosystem. It focuses on seamless consumer experiences enabling everyone and everything connected to the Internet to contribute to satisfying such personalized demands. Secondly, wemarket considers all participants as equal in their rights and responsibilities as they can engage directly without any intermediaries and related constraints. They are constitutive parts not only through the intention to participate in market exchange but also through their intention to provide for the underlying infrastructure and market mechanisms. This case study considered the context of a smart city. However, the same holds true for more significant areas like smart regions and, ultimately, non-geographical areas, i.e., market spaces on Internet scale.

**Analysis:** The proposed reference model supports a structured analysis of aspects of the strategic and operational levels of a decentralized and self-organized online structure. As a tool for analysis instruments, it separates concerns of these two levels guiding the analysis through three dimensional phases, views, and stages. Each of these dimensions has a different focus and compounds of different elements that need to be analyzed separately in order to gain a comprehensive overview. The proposed framework, therefore, assists in gaining a deeper understanding of relevant entities, elements, and relationships between them, and thus facilitates defining the project scope and its resulting design requirements. Furthermore, it supports an early estimation of competencies considered essential for design activities, and, eventually, implementation (i.e., competencies necessary for modeling interaction processes (interaction view) or IT expertise inherent for activities associated with services and infrastructure). In the wemarket project, these competencies were assumed to exist, but this does not necessarily need to be the case. Therefore, training the project team in the application of the reference model for distributed market spaces is highly recommended in order for the team to use it adequately and beneficially.

**Design:** As a design instrument, the proposed reference model structures the design process providing a set of four steps that are organized cyclically, thus constituting a wheel model. This ranges from the process of the modeling of ecosystem structure over the core interactions and services to their prototypical implementation. Each step relies on the inputs of the previous one, and each step produces a clearly defined outcome in the form of derived models. Together these four models compose a blueprint that blends the results of different design activities on the one hand and integrates the application context on the other hand. That ensures that all models are derived in context and serve the defined value proposition. In this work, we presented only one iteration of the blueprinted models, as the focus was to implement an instance of wemarket.space. This instance should be seen as one of the first iterations of the design process. As to the applied wheel model (cf. Figure 19), this first instance should be seen as the test system required to test the blueprinted models rather than a final sample of the system.

**Implementation:** Most examined reference models and approaches (cf. Section 3) focus solely on modeling activities, leaving the prototyping to the software developer responsible for the implementation. Our proposed model goes a step further proposing an accompanying architecture for distributed market spaces [8,9] as the possible implementation of the infrastructure view. Hence it supports the implementation of a reference DMS on the operational level. Therefore the proposed reference model comes with additional advantages. Firstly, it speeds up the prototyping of the blueprinted models, given the detailed description of the underlying software system. Due to the specified functional and information structure of the system, the prototyping activities can focus on the realization of the proposed solution and thus lead to more reliable results in a shorter period. For the prototyping of the wemarket project (cf. Section 4.5), Web technologies have been used. However, the final choice of technologies is subject to a concrete implementation for a particular application context and depends on the team's competences, as mentioned before. Secondly, it enables an early evaluation of the design hypothesis and associated design decisions. The blueprinted structure incorporates design decisions that are made based on the understanding and assumption of the application's context and associated requirements. Since these assumptions might not be complete or even valid, the prototyped system will realistically validate the results of the design activities. As a result, possible deficiencies are identified, and modifications defined regarding the ecosystem structure, interactions, or service stack. Such a feedback loop allows the blueprint to undergo necessary iterations in order to arrive at the state of the so-called "minimum viable product". That means to a minimum viable system necessary to operate in order for the defined value proposition to be released.

The proposed reference model concentrates on different aspects of distributed market spaces and their inner-workings on the strategic and operational level, which are relevant for market transactions for complex products, as well as on how its instances might unfold during different life stages. However, our reference model disregards aspects related to regulatory affairs. These are particularly important for market transactions of complex products that are realized in a cross-border transaction (i.e., when providers from different countries are involved). Currently, such transnational regulations, especially in online trading, differ from one country to another. They might even differ from region to region, or might form part of current political considerations as is the case with Single Market Initiative of the European Commission [67]. Therefore, such regulatory constraints need to be considered as part of the contextual requirements, and that they can become part of design decisions.

Moreover, the proposed reference model disregards concepts that deal with the identity and privacy of users (i.e., actors). Our reference model draws on the positive collective motivation, yet, forgery and fraud are undesired practices that any market-oriented environment faces to some extent. There are different approaches to this issue, e.g., NICE [68] and SOLID [69], that can be taken into consideration to prevent or at least mitigate undesired behavior.

Furthermore, some sensibilities and trade-offs derive from the strictly decentralized technical infrastructure (i.e., software-system for the implementation of the infrastructure view). These are sensitivities linked to decentralized governance and data management and viewed as the main trade-offs to alleviate the adverse effects of the positional power of centrally governed environments. Nevertheless, these are out of the scope of this work and subject to future work.

## 7. Conclusions and Future Work

In this paper, we introduced our Reference Model for Distributed Market Spaces. It is a multi-dimensional and multi-view model for the analysis, design, and implementation of decentralized and self-organized online structures.

As the theoretical framework, our reference model leveraged the St. Gallen Media Reference Model. It extended and adjusted the model by Schmid and Lindemann to meet the overall objectives of a reference DMS, and hence integrated additional elements necessary to cope with market transactions for complex products. The proposed model spans three dimensions defining a reference DMS in regards to Views, Phases, and Stages. The Views dimension describes a reference DMS in its organizational

structure (ecosystem view). It specifies the core interactions among the involved participants (interaction view) and defines services that a reference DMS must provide to its participants (service view). Additionally, it specifies the technical infrastructure, which is necessary for the implementation of the specified services (infrastructure view). The Phases dimension defines a reference DMS as a market-oriented environment to support transactions of complex products. It is comprised of the knowledge, intention, contract, settlement, and follow-up phases, which determine how to initiate, arrange and settle contractual agreements for complex products in a way as to lowering transaction costs for consumers looking for such products. The Stages dimension considers the life stages a reference DMS might undergo during its development and growth: design, ignition, and maturity stages. Together, the views, phases, and stages describe how a reference DMS works on the strategic and operational levels, how it enables market transactions for complex products and, how its instances might unfold during different life stages.

Our reference model was applied and evaluated in a case study on smart cities as part of the wemarket project. The outcomes of the conducted case study demonstrated how an instance of wemarket had been blueprinted, prototyped, and launched as a demonstrator. Therefore, we argue that our reference model meets the main objectives it was designed for. However, some aspects were identified, which are currently not considered by our proposed model. These relate to regulatory elements, which are relevant for market transactions of complex products realized transnationally. The same holds true for the identity and privacy elements, deemed crucial to distributed market spaces in decentralized and self-organized environments distributed market spaces. These require further studies and will be the subject of our future work.

**Author Contributions:** M.R.-S. worked on the formal analysis, methodology, conceptualization, prototype development, validation, and writing original draft preparation as well as editing. D.P. worked on the analysis, validation and project administration.

**Funding:** This research received no external funding.

**Acknowledgments:** This work is kindly supported by Baden-Wuerttemberg Cooperative State University Mannheim, Digital Transformation Center (DTC), and Department of Business Informatics.

**Conflicts of Interest:** The authors declare no conflict of interest.

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
