# Peer review of "Beyond Platform Economy: A Comprehensive Model for Decentralized and Self-Organizing Markets on Internet-Scale"

_computers, doi:10.3390/computers8040090_

Round 1
Reviewer 1 Report
Dear,
This is an interesting and dared article. Provides a new framework and a value experience in the test of it. Despite that, in my opinion it has two main lacks that can improve its quality (in terms of an academic paper). On the one hand, some statements should be better argued/referenced. On the other hand, there are some used concepts/expressions that could be considered.
Taking advantage of the message I am attaching the manuscript with some comments to illustrate what I am trying to transmit.
In any case, I'm encouraging the author to take into account these considerations in order to publish this interesting paper.

Author Response
Dear Reviewer
Thank you very much for your valuable feedback, based upon which we revised our manuscript.
We have addressed the majority of your suggestions by adding additional references and more details related to commented statements. Our changes refer in particular to Sections 1. Introduction, 2. Background and Approach, and 3. Related Work.
Please find attached the new version of our manuscript – changes are highlighted.
About your comment on pg. 10:
"Knowledge: In my point of view, knowledge is a relevant word here. Maybe something like "content" suit better?"
The original MRM proposed the name of the Phase "Knowledge"; so we leave it that way. But in our further work, we will consider changing it in "Content".
Yours Sincerely
Authors

Reviewer 2 Report
First of all, I would like to state that this is a very good paper and one can see that there is a lot of effort included in it.
Figure 2 - title should be corrected - complex should be instead cmplex
The discussion part can be improved by elaborating more on the case study and results of the case study because now the theoretical part is taking most of the paper while the research part (case study) is not used enough for this paper. This is a suggestion which I believe will improve further the paper.
Author Response
Dear Reviewer
Thank you very much for your valuable feedback, based upon which we revised our manuscript.
We have addressed the majority of your suggestions by adding additional references and more details related to commented statements. Our changes refer in particular to Sections 1. Introduction, 2. Background and Approach, and 3. Related Work as well as 6. Discussion.
Please find attached the new version of our manuscript – changes are highlighted.
About your comment, "The discussion part can be improved by elaborating more on the case study and results of the case study because now the theoretical part is taking most of the paper while the research part (case study) is not used enough for this paper.?"
The submitted manuscript was generated using the LaTEX tool, which did not compile all sections as intended by the authors. The question mark in the Introduction (last paragraph) indicated that something went wrong during the compiling process. As a result, Section 4 "Proposed Reference Model for Distributed Market Spaces" appears as a part of Section 3 "Related Work" which provided a wrong impression that the main contribution of the manuscript was the conducted case study instead of proposing a reference model for distributed market spaces. The case study was conducted to demonstrate the applicability of the proposed reference model in a specific application context (in this case, in the context of the wemarket project).
Despite that, we agree that the results of the case study could be more discussed. Therefore, we added additional remarks, which (we do hope) will clarify the goal and the scope of the case study. Accordingly, the case study was used to demonstrate how the proposed reference model could be applied in the context of the wemarket project. In this case, the focus was on establishing the instance of the wemarket.space and using it for the testing of blueprinted models (ecosystem, interactions, services, and infrastructure).
In addition, we provided online resources that show sample resources and demonstrate the functionality of an instance of a wemarket.space in real-life modus.
Yours Sincerely
Authors

Reviewer 3 Report
Dear authors,
I have read and analyzed your article and I have the following remarks:
- The article is well structured in terms of introduction, literature review and proposed model. However, the proposed model is a theoretical one and the case study that applies it is, in my opinion far too simplistic, compared to the presumption from which the entire article starts: that the proposed model is an alternative to platform economy with global platform such as Amazon or Alibaba.
- The literature review regarding the reference models classes is comprehensive and well structured but well too extensive giving the reader difficulties to follow the red line of the article
- Theorizing the platform models is anyway a type of "reverse engineering" because, as business models, they appeared in practice independent of the scientific researches that subsequently theorized them.
- The model proposed as a post-platform economy model to decentralize this business model has, in my opinion, some vulnerable points especially regarding the possibility for the actors to play multiple roles which would ultimately lead to the consolidation of the position of some of them and they will thus be able to take control at least partially. Given the fact that apart from the consumer, all the other actors aim to increase their competitive advantage with the final goal of obtaining a profit, I do not see how the proposed model can offer an alternative to this behavior. Existing platforms in the global economy show that they have gone from vertical development to the inclusion of more verticals since they have developed an infrastructure to support the inclusion of more independent businesses with lower transaction costs.
- The case study offered is, on the other hand, too simplistic for the multitude of aspects involved in such a platform and practically contravenes with the researches on consumers' behavior who are currently pursuing their needs at a higher level seeking the purchase of a service by investigating as much as possible details and personalized aspects.
In conclusion, I believe that the article fulfills the formal conditions of a scientific article from the point of view of the structure but did not convince me from the point of view of a viable scientific research result for the real economy.
The use of English language
check the text for small mistakes as missing letters or wrong letters may be it will be a plus to verify the abusive use of adjectives and adverbsAuthor Response
Dear Reviewer
Thank you very much for your valuable feedback, based upon which we revised our manuscript.
Please find attached the new version of our manuscript – changes are highlighted.
In the following, we address your comments and discuss our changes to the paper in response to each comment. If we did not change the manuscript based on a particular suggestion or feedback, we would explain the rationale behind it.
"- The article is well structured in terms of introduction, literature review and proposed model. However, the proposed model is a theoretical one and the case study that applies it is, in my opinion far too simplistic, compared to the presumption from which the entire article starts: that the proposed model is an alternative to platform economy with global platform such as Amazon or Alibaba.
- The literature review regarding the reference models classes is comprehensive and well structured but well too extensive giving the reader difficulties to follow the red line of the article
- Theorizing the platform models is anyway a type of "reverse engineering" because, as business models, they appeared in practice independent of the scientific researches that subsequently theorized them.?"
The submitted manuscript was generated using the LaTEX tool, which did not compile all sections as intended by the authors. The question mark in the Introduction (last paragraph) indicated that something went wrong during the compiling process. As a result, Section 4 "Proposed Reference Model for Distributed Market Spaces" appears as a part of Section 3 "Related Work" which provided a wrong impression that the main contribution of the manuscript was the conducted case study instead of proposing a reference model for distributed market spaces. The case study was conducted to demonstrate the applicability of the proposed reference model in a specific application context (in this case, in the context of the wemarket project).
Despite that, we agree that the case study, we used in this work is simplistic and that its results could be more discussed. Therefore, we added additional remarks, which (we do hope) will clarify the goal and the scope of the case study. Accordingly, the case study was used to demonstrate how the proposed reference model could be applied in the context of the wemarket project. In this case, the focus was on establishing the instance of the wemarket.space and using it for the testing of blueprinted models (ecosystem, interactions, services, and infrastructure).
In addition, we provided online resources that show sample resources and demonstrate the functionality of an instance of a wemarket.space in real-life modus.
Furthermore, we will use the instantiated wemarket-space as the grounds for further studies. In particular, for testing the hypothesis made about the "enabler roles" and their motivation to participate and provide for the ecosystem. Some projects and concepts around "platform cooperativism" (Scholz, 2016) and related works - which we added to Section 3 - can offer more insights and will be as subject to our future work.
"-The use of English language: check the text for small mistakes as missing letters or wrong letters may be it will be a plus to verify the abusive use of adjectives and adverbs"
Some apparent mistakes and missing letters ware corrected. A professional language service will revise the final version of the manuscript.
Yours Sincerely
Authors
